# Technical note: Estimating aqueous solubilities and activity coefficients of mono- and $\alpha,\omega$-dicarboxylic acids using COSMO*therm*

Noora Hyttinen[1,2], Reyhaneh Heshmatnezhad[1], Jonas Elm[3], Theo Kurtén[2], and Nønne L. Prisle[1]

[1]Nano and Molecular Systems Research Unit, University of Oulu, P.O. Box 3000, FI-90014 Oulu, Finland
[2]Department of Chemistry and Institute for Atmospheric and Earth System Research (INAR), University of Helsinki, P.O. Box 55, FI-00014 Helsinki, Finland
[3]Department of Chemistry and iClimate, Aarhus University, Langelandsgade 140, 8000 Aarhus C, Denmark

**Correspondence:** Noora Hyttinen (noora.hyttinen@oulu.fi), Nønne L. Prisle (nonne.prisle@oulu.fi)

**Abstract.** We have used the COSMO*therm* program to estimate activity coefficients and solubilities of mono- and $\alpha,\omega$-dicarboxylic acids, and water in binary acid–water systems. The deviation from ideality was found to be larger in the systems containing larger acids than in the systems containing smaller acids. Conductor-like Screening Model for Real Solvents (COSMO-RS) underestimates experimental monocarboxylic acid activity coefficients by less than a factor of 2 but experimental water activity coefficients are underestimated more especially at high acid mole fractions. We found a better agreement between COSMO*therm*-estimated and experimental activity coefficients of monocarboxylic acids when the water clustering with a carboxylic acid and itself was taken into account using the dimerization, aggregation and reaction extension (COSMO-RS-DARE) of COSMO*therm*. COSMO-RS-DARE is not fully predictive, but fit parameters found here for water–water and acid–water clustering interactions can be used to estimate thermodynamic properties of monocarboxylic acids in other aqueous solvents, such as salt solutions. For the dicarboxylic acids, COSMO-RS is sufficient for predicting aqueous solubility and activity coefficients and no fitting to experimental values is needed. This is highly beneficial for applications to atmospheric systems, as these data are typically not available for a wide range of mixing states realized in the atmosphere, due to lack of either feasibility of the experiments or of sample availability. Based on effective equilibrium constants of different clustering reactions in the binary solutions, acid dimer formation is more dominant in systems containing larger dicarboxylic acids (C5-C8), while for monocarboxylic acids (C1-C6) and smaller dicarboxylic acids (C2-C4), hydrate formation is more favorable, especially in dilute solutions.

## 1  Introduction

Mono- and dicarboxylic acids ($CH_3(CH_2)_{n-2}COOH$ and $COOH(CH_2)_{m-2}COOH$, respectively) are common atmospheric compounds that have been detected in both the gas (Kawamura et al., 2000; Fisseha et al., 2006) and aerosol phase (Fisseha et al., 2006; Verma et al., 2017; Guo et al., 2014; Hyder et al., 2012). Carboxylic acids have been detected in high abundance in various environments, such as urban (Zhao et al., 2018; Kawamura et al., 2000; Fisseha et al., 2006; Jung et al., 2010; Guo et al., 2014), semi-urban (Verma et al., 2017), marine (Kawamura and Sakaguchi, 1999; Mochida et al., 2003) and Antarctic

(Kawamura et al., 1996) measurement sites. In general, small carboxylic acids ($n \leq 3$ and $m \leq 4$) are more abundant than large acids ($n > 3$ and $m > 4$) (Jung et al., 2010; Fisseha et al., 2006; Tsai and Kuo, 2013; Zhao et al., 2018; Guo et al., 2014; Kawamura et al., 2000). For example, Tsai and Kuo (2013) found that 77.2% of all carboxylic acids in fine particulate matter ($PM_{2.5}$) were small carboxylic acids (formic, acetic and oxalic acid) in a broad-leaved forest in central Taiwan.

Accurate description of the different aerosol phases is important for determining parameters used in aerosol modeling, such as gas-to-particle partitioning, in particular water uptake and chemical reactivity. A large number of reactions in the aqueous aerosol phase are strongly pH dependent (Pye et al., 2020; Weber et al., 2016), but accurate predictions of aerosol acidity are highly challenging. One element to resolve is the nature and amount of acidic material dissolved in the aqueous aerosol phase. The aqueous bulk solubility of mono- and dicarboxylic acids have been measured in multiple studies (Saxena and Hildemann, 1996; Apelblat and Manzurola, 1987, 1989, 1990; Cornils and Lappe, 2000; Song et al., 2012; Romero and Suárez, 2009; Omar and Ulrich, 2006; Brooks et al., 2002). However, acid activity data of carboxylic acid–water systems is much scarcer. Jones and Bury (1927) derived the activity coefficients of formic ($n = 1$), acetic ($n = 2$), propanoic ($n = 3$) and butanoic ($n = 4$) acids in aqueous solutions at the freezing points of the binary solutions using freezing point depression measurements. Using freezing point depression measurements, activity coefficients are calculated using Lewis and Randall's equation for non-electrolytes. Hansen et al. (1955) derived activity coefficients of acetic, propanoic and butanoic acid in water and the activity coefficients of water in acetic, propanoic, butanoic, pentanoic ($n = 5$) and hexanoic ($n = 6$) acids, at 298.15 K, using partial pressure measurements. In addition, Hansen et al. (1955) represented the experimental points using self-consistent activity coefficient functions. Activity coefficients of malonic, succinic and glutaric acid ($m = 3$, 4 and 5) have been measured by Davies and Thomas (1956) and Soonsin et al. (2010) in bulk and particle experiments, respectively.

Group contribution methods, such as UNIFAC (Fredenslund et al., 1975) and AIOMFAC (Zuend et al., 2008), are often used to estimate activity coefficients of atmospherically relevant compounds. More recently, a quantum chemistry based Conductor-like Screening Model for Real Solvents (COSMO-RS: Klamt, 1995; Klamt et al., 1998; Eckert and Klamt, 2002) has been used to predict thermodynamic properties of multifunctional compounds. Solubilities and activity coefficients of carboxylic acids have also been estimated using the COSMO-RS theory implemented in the COSMO*therm* program (COSMO*therm*, 2019). For instance, Schröder et al. (2010) estimated the aqueous solubilities of various polycarboxylic acids using the TZVP parametrization of COSMO*therm* and found that COSMO*therm* was able to predict the temperature dependence of the solubilities of dicarboxylic acids ($m = 2$–8) well, while the absolute solubility estimates were not in a good agreement with experiments. Additionally, Michailoudi et al. (2020) estimated the activity coefficients of monocarboxylic acids with even number of carbon atoms ($n = 2, 4, 6, 8, 10, 12$) at infinite dilution. In addition, they estimated the solubility of the same acids in pure water and different aqueous electrolyte solutions. They found a good agreement between experimental and estimated aqueous solubilities of the acids with the exception of butanoic acid, which in experiments has been seen to be fully soluble (Saxena and Hildemann, 1996), while COSMO*therm* predicted a finite solubility.

Recent work has shown that the absolute COSMO*therm* solubility and activity coefficient estimates can be improved by excluding conformers containing intramolecular hydrogen bonds from the COSMO*therm* calculation (Hyttinen and Prisle, 2020). However, based on the hydrogen bonding definition of COSMO*therm*, monocarboxylic acids are not able to form

intramolecular hydrogen bonds. Therefore, other methods are needed to improve COSMO*therm* estimates of monocarboxylic acids. On the other hand, carboxylic acids are able to form hydrogen bonded dimers where two molecules are bound by two simultaneous intermolecular hydrogen bonds. These concerted multiple contacts, such as is seen in carboxylic acid dimer formation, are not captured by COSMO-RS. A dimerization, aggregation and reaction extension to the COSMO-RS theory (COSMO-RS-DARE) was developed to account for these interactions (Sachsenhauser et al., 2014). For example, Cysewski (2019) was able to improve the agreement between experimental and estimated solubilities of ethenzamide in various organic solvents using COSMO-RS-DARE.

Most atmospherically relevant multifunctional compounds are not readily available for experimental determination of thermodynamic properties. Accurate theoretical estimates are therefore essential for advancing current aerosol process modeling to include more complex compounds and mixtures. Here, we demonstrate the applicability of COSMO-RS theory in calculating condensed-phase properties of atmospherically relevant organic compounds. Carboxylic acids are among the most abundant and well characterized organic compounds in the troposphere and are therefore a good compound class to use to validate the use of COSMO-RS in atmospheric research. We use the newly developed COSMO-RS-DARE, as well as COSMO-RS, to estimate activity coefficients of monocarboxylic acids ($n = 1$–$6$) and $\alpha,\omega$-dicarboxylic acids ($m = 2$–$8$), and water, in binary acid–water mixtures. In addition, we estimate aqueous solubilities and effective equilibrium constants of cluster formation of the acids.

## 2   COSMO*therm* calculations

We use the COSMO*therm* software (release 19 and parametrization BP_TZVPD_FINE_19) (COSMO*therm*, 2019) to estimate the solubilities and activity coefficient of linear mono- and dicarboxylic acids in binary aqueous solutions. In addition, we compute the effective equilibrium constants of water and acid dimerization (formation of a hydrogen bonded cluster containing two water molecules or two acid molecules, respectively), and acid hydration (formation of a hydrogen bonded cluster containing one acid and one water molecule).

### 2.1   Activity coefficients

COSMO*therm* calculates the activity coefficient ($\gamma$) of compound $i$ with mole fraction $x_i$ using the pseudo-chemical potentials at composition $\{x_i\}$ ($\mu_i^*(x_i)$) and at the reference state ($\mu_i^{*\circ}(x^\circ, T, P)$). By default, the reference state used in COSMO*therm* is the pure compound (labeled as convention I; Levine, 2009):

$$\ln \gamma_i^{\mathrm{I}}(x_i) = \frac{\mu_i^*(x_i) - \mu_i^{*,\mathrm{I}\circ}(x^\circ, T, P)}{RT} \tag{1}$$

at $P = 10^5$ Pa reference pressure. $T$ is the temperature (K) and $R$ the gas constant (kJ K$^{-1}$ mol$^{-1}$, when $\mu^*$ is given in kJ mol$^{-1}$).

Pseudo-chemical potential (Ben-Naim, 1987) is an auxiliary quantity defined using the chemical potential at the reference state $\mu^\circ$:

$$\mu_i^*(x_i) = \mu_i{}^\circ(x^\circ, T, P) + RT \ln \gamma_i(x_i) \tag{2}$$

Pseudo-chemical potential has recently been used in molecular level solvation thermodynamics instead of chemical potential (Sordo, 2015). The benefit of pseudo-chemical potential is that it is valid for any concentration and fluid mixture, while the conventional chemical potential cannot necessarily be used to describe infinite dilution ($x_i \to 0$)(Ben-Naim, 1978). By definition, the activity coefficient of a compound at the reference state is unity ($\gamma_i^{\mathrm{I}}(x_i = 1) = 1$), which leads to equal chemical and pseudo-chemical potential at the reference state. At other states ($x_i < 1$), the relation between chemical and pseudo-chemical potentials ($\mu$ and $\mu^*$, respectively) can be expressed as

$$\mu_i^*(x_i) = \mu_i(x_i) - RT \ln x_i \tag{3}$$

Unless otherwise mentioned, the mole fractions $x_i$ correspond to mole fractions of undissociated acid or neutral non-protonated water.

## 2.2 Solubility

Solubilities are calculated by finding the liquid-liquid (LLE) or the solid-liquid equilibrium (SLE) of the binary liquid–water or solid–water systems, respectively. In LLE, the chemical potential ($\mu$) of a compound is equal in both of the liquid phases ($\alpha$ and $\beta$):

$$\mu_i^\alpha = \mu_i^\beta \tag{4}$$

Combining equations 2 and 3 gives the relation between chemical potential and pseudo-chemical potential at the reference state:

$$\mu_i(x_i) = \mu_i^{*\circ}(x^\circ, T, P) + RT \ln x_i + RT \ln \gamma_i(x_i) \tag{5}$$

Equation 5 can be substituted for chemical potential in equation 4, giving

$$\mu_i^{*\circ}(x^\circ, T, P) + RT \ln a_i^\alpha = \mu_i^{*\circ}(x^\circ, T, P) + RT \ln a_i^\beta, \tag{6}$$

where $a_i^\alpha$ and $a_i^\beta$ are the activities ($a = x\gamma$) of compound $i$ in phases $\alpha$ and $\beta$, respectively. The liquid-liquid equilibrium condition between the solvent-rich phase ($\alpha$) and the solute-rich phase ($\beta$) becomes:

$$a_i^\alpha = a_i^\beta \tag{7}$$

The SLE is solved from the solid-liquid equilibrium condition (Eckert and Klamt, 2019):

$$\log_{10}(x_{\mathrm{SOL},i}) = \frac{\mu_i^{*,\mathrm{I}\circ}(x^\circ, T, P) - \mu_i^*(x_{\mathrm{SOL},i}) - \Delta G_{\mathrm{fus}}(T)}{RT \ln(10)}, \tag{8}$$

where $x_{\text{SOL},i}$ is the mole fraction solubility (SOL) of compound $i$ in the solvent. The free energy of fusion of the solute ($\Delta G_{\text{fus}}(T)$) is calculated from the experimentally determined heat of fusion ($\Delta H_{\text{fus}}$) and melting point ($T_{\text{melt}}$) using the Schröder-van Laar equation (Prigogine and Defay, 1954):

$$\Delta G_{\text{fus}}(T) = \Delta H_{\text{fus}}(1 - \frac{T}{T_{\text{melt}}}) - \Delta C_{p,\text{fus}}(T_{\text{melt}} - T)$$
$$+ \Delta C_{p,\text{fus}} T \ln \frac{T_{\text{melt}}}{T} \tag{9}$$

Here the heat capacity of fusion ($\Delta C_{p,\text{fus}}$) is estimated from the melting point and the heat of fusion:

$$\Delta C_{p,\text{fus}} = \frac{\Delta H_{\text{fus}}}{T_{\text{melt}}} \tag{10}$$

Table 1 shows experimental melting points and heats of fusion of the dicarboxylic acids of this study. Melting points and heats of fusion of the monocarboxylic acids are not used, since all of the monocarboxylic acids studied here are in liquid phase at 298.15 K.

## 2.3 Effective equilibrium constants

COSMO*therm* estimates effective equilibrium constants of condensed-phase reactions from the free energy of the reaction ($\Delta G^{\text{I}\circ}{}_r$):

$$K_{\text{eff}} = e^{-\frac{\Delta G^{\text{I}\circ}{}_r}{RT}} \tag{11}$$

The reaction free energy is calculated from the free energies of the pure reactants ($G^{\text{I}\circ}{}_{\text{react}}$) and products ($G^{\text{I}\circ}{}_{\text{prod}}$):

$$\Delta G^{\text{I}\circ}{}_r = \Sigma G^{\text{I}\circ}{}_{\text{prod}} - \Sigma G^{\text{I}\circ}{}_{\text{react}} \tag{12}$$

The free energy of compound $i$ is the sum of the energy of the solvated compound ($E_{\text{COSMO}}$), the averaged correction for the dielectric energy ($dE$; Klamt et al., 1998) and the pseudo-chemical potential of the pure compound:

$$G^{\text{I}\circ}{}_i = E_{\text{COSMO},i} + dE_i + \mu_i^{*,\text{I}\circ}(x^\circ, T, P) \tag{13}$$

## 2.4 Concentration dependent reactions (COSMO-RS-DARE)

In COSMO-RS, the surface of a molecule is divided into surface segments that represent the surface charges of the molecule. The surface is considered as an interface between a virtual conductor around the molecule and the cavity formed by the molecule (Klamt and Schüürmann, 1993). Each surface segment has an area ($[\text{Å}^{-2}]$) and a screening charge density ($\sigma[\text{e Å}^{-2}]$). Interactions between molecules are described through the interaction between surface segments of the different molecules. Examples of $\sigma$-surfaces used in COSMO*therm* calculations are shown in Fig. 1. The red color of a $\sigma$-surface signifies a positive screening charge density (negative partial charge) and the blue color a negative screening charge density (positive partial charge).

Concerted multiple contacts, such as carboxylic acid dimer formation, are not captured by COSMO-RS. COSMO*therm* is able to consider these hydrogen bonded clusters using the dimerization, aggregation, and reaction extension (COSMO-RS-DARE, Sachsenhauser et al. (2014)). We use the COSMO-RS-DARE method in our activity coefficient and solubility calculations. In our equilibrium constant calculations, the clusters in the system are included as the product of the clustering reactions. The method is described below and the full COSMO-RS-DARE derivation can be found from Sachsenhauser et al. (2014).

A clustering reaction between molecules A and B can be described by the equilibrium:

$$A + B \rightleftharpoons A \cdot B \tag{R1}$$

In acid–water systems, A and B can be either a carboxylic acid or a water molecule. In COSMO-RS-DARE, the product clusters are included in COSMO*therm* calculations by using the $\sigma$-surfaces of molecule A in the cluster, omitting the part of the $\sigma$-surface that is assigned to the molecule clustered with A (i.e., molecule B). Similarly, the clustering product of molecule B is included in the calculation by omitting the $\sigma$-surface assigned to molecule A from the $\sigma$-surface of A·B. Examples of these partial $\sigma$-surfaces are shown on the right hand side of Fig. 1.

The formation of hydrogen bonds (in hydrates or dimers) is taken into account using the interaction energy of the two reacting compounds. The formation free energy of the cluster ($G(A,A \cdot B)$) is calculated using fit parameters $c_H$ and $c_S$ (enthalpic and entropic contributions, respectively) to describe the interaction between the monomers A and B in the cluster (A·B):

$$G(A, A \cdot B) = c_H - c_S T, \tag{14}$$

The fit parameters are used because COSMO*therm* is unable to calculate the energy of a monomer in a cluster. Instead, the energy of a monomer in a cluster is assumed to be equal to the energy of the lowest-energy conformer of the same compound and the favorability of the cluster formation is estimated using the fit parameters. Without temperature dependent experimental data, it is not possible to fit both fit parameters. We therefore consider the enthalpic parameter $c_H$ as the total formation free energy parameter at 298.15 K, setting the entropic parameter $c_S$ to zero.

COSMO-RS-DARE was originally developed for systems containing carboxylic acids in non-polar solvents (Sachsenhauser et al., 2014). In a carboxylic acid–water system, both the carboxylic acid and water are able to form strongly bound clusters. In addition, hydrated acids can be formed. We are thus including the interactions of the clustering reactions for both A and B, even when A = B.

## 2.5 Input file generation

The .cosmo files of water and the monocarboxylic acids with a low number of conformers ($<10$) are taken from the COSMO*base* (COSMO*base*, 2011) database. For the dicarboxylic acids, acid and water dimers, and the hydrates of pimelic ($m = 7$) and suberic ($m = 8$) acids, we use the following systematic conformer search approach detailed by Kurtén et al. (2018) as it has been shown to give more consistent results than other conformer sampling approaches. The conformers are found using the systematic conformer search in the Spartan program (Wavefunction Inc., 2014, 2016). The conformer set is then used as input

to the COSMO*conf* program (COSMO*conf*, 2013) (using the TURBOMOLE program (TURBOMOLE, 2010)), which runs initial single-point COSMO calculations at the BP/def-SV(P) level of theory to compare the pseudo-chemical potentials of the conformers and remove similar structures. Initial geometry optimizations are calculated at the BP/def-SV(P) level of theory, duplicate structures are removed by comparing the new geometries and pseudo-chemical potentials. Final geometries are optimized at the BP/def-TZVP level of theory and after a second duplicate removal step, final single-point energies are calculated at the BP/def2-TZVPD-FINE level of theory.

For acid dimers, we use the lowest gas-phase energy structures found by Elm et al. (2019) as a starting structure for systematic conformer search. For hydrated monocarboxylic acids and smaller dicarboxylic acids ($m \leq 6$), the clusters are built by adding a water molecule to each conformer of the free acids. For monocarboxylic acids, the water molecule is placed on the carboxylic acid group forming two intermolecular hydrogen bonds between the molecules. For dicarboxylic acids, a water molecule is added to either end of the acid, forming two hydrate conformers from a single acid conformer. For the dicarboxylic acid conformers with the two acid groups close to each other, additional conformers are created for cases where the water molecule is interacting with both acid groups. Fig. 2 illustrates the formation of two different adipic acid hydrate conformers from a single monomer conformer. A cluster conformer where the water molecule is attached to one carboxylic acid group is shown on the top right corner and in the bottom right corner conformer, the water molecule is bound to both acid groups. Due to the large number of conformers of non-hydrated pimelic ($m = 7$) and suberic ($m = 8$) acid (75 and 132, respectively), the monohydrate conformers of those two acids are sampled separately using Spartan.

We use only clusters of two molecules in our calculations. In carboxylic acid dimers, the hydrogen bond donors and acceptors are saturated, which means that carboxylic acids are unlikely to form larger clusters than dimers (Vawdrey et al., 2004; Elm et al., 2014, 2019). Computational studies (Aloisio et al., 2002; Weber et al., 2012; Kildgaard et al., 2018) have shown that, in the gas phase, the energetically most favorable dihydrate is formed by two water molecules attaching to the same carboxylic acid group. Therefore, adding a second water molecule to the cluster does not significantly change the probability distribution of the screening charge density ($\sigma$-profile) of the acid in the cluster compared to the acid in a monohydrate or dimer.

Conformers containing no intramolecular hydrogen bonds (Kurtén et al., 2018; Hyttinen and Prisle, 2020) are used in the COSMO-RS solubility and activity coefficient calculations. Due to the intermolecular hydrogen bonding in the hydrate and dimer clusters, all conformers (up to 40 conformers) of monomers and clusters are used in the effective equilibrium constant calculations. In COSMO-RS-DARE calculations, we use all conformers of the monomers and only the lowest solvated energy conformers of the clusters.

## 3  Results and discussion

### 3.1  Effective equilibrium constants of clustering reactions

We estimated the effective equilibrium constants of the different clustering reactions (i.e., hydration and dimerization) of the binary acid–water systems. A comparison between the hydration and acid dimerization equilibrium constants in the aqueous phase is given in Fig. 3 and Table 2. The equilibrium constants for both the dimerization and hydration reactions are similar

between all of the monocarboxylic acids, and are not labeled in Fig. 3. For the dicarboxylic acid, we can see larger variation in both the hydration and dimerization reactions. Note that the COSMO-RS-DARE method is not used in the effective equilibrium constant calculations because the clusters are already included in the calculation as products.

For all of the acids, the effective equilibrium constant of dimerization is higher than that of the hydrate formation of the corresponding acid, meaning that acid dimer formation is energetically more favorable than hydrate formation. However, in dilute conditions, water is more abundant, shifting the equilibrium from acid dimerization to hydration. The dimerization:hydration ratio is the lowest for oxalic ($m = 2$) and malonic ($m = 3$) acids, while monocarboxylic acids and succinic acid ($m = 4$) have similar (intermediate) ratios, and the larger dicarboxylic acids ($m = 5$–$8$) have higher ratios. This means that, in dilute solutions, oxalic, malonic and succinic acid will most likely interact with water instead of other acid molecules.

Vawdrey et al. (2004) calculated the dimerization enthalpies (at the B3LYP/6-31++G(2d,p) level of theory) of monocarboxylic acids ($n = 2$–$6$) and found a notable even–odd variation (dimerization of the acids with odd number of carbon atoms is more favorable than of the even carbon number acids). The same is seen here in the condensed phase, where the effective equilibrium constant of butanoic and hexanoic acids are lower than of propanoic and pentanoic acids, respectively. Otherwise there is a slightly increasing trend in the effective equilibrium constants with the increasing number of carbon atoms in the monocarboxylic acids. For larger dicarboxylic acids ($m \geq 4$), Elm et al. (2019) found an even–odd alternation in the dimer:monomer ratio in the gas phase, calculated at the DLPNO-CCSD(T)/aug-cc-pVTZ//$\omega$B97X-D/6-31++G(d,p) level of theory. We observe a similar increasing effective equilibrium constant with the increasing carbon chain length in the smaller dicarboxylic acids ($m = 2$–$5$) and an even–odd alternation in the larger dicarboxylic acids ($m = 4$–$8$) in any condensed phase.

## 3.2 Activity coefficients

### 3.2.1 Monocarboxylic acids

We calculated the activity coefficient of the carboxylic acids and water in the binary acid–water mixtures using the COSMO-RS-DARE method. Hansen et al. (1955) derived the activity coefficients of acetic ($n = 2$), propanoic ($n = 3$) and butanoic ($n = 4$) acids in mixtures with water from partial pressure measurements. In addition, they determined activity coefficients of water in aqueous acetic, propanoic, butanoic, pentanoic ($n = 5$) and hexanoic ($n = 6$) acid mixtures. We used these experimental activity coefficients to fit the enthalpic parameters ($c_H$) for each of the acids in the COSMO-RS-DARE calculations. Figure 4 shows a comparison between the estimated and experimentally determined activity coefficients of these monocarboxylic acids, and formic acid ($n = 1$), for which no experimental activity coefficient data is available.

The reactions included in the calculations are water dimer ($H_2O \cdot H_2O$) and acid hydrate ($RCOOH \cdot H_2O$) formation. A comparison between COSMO-RS activity coefficients and the COSMO-RS-DARE method with different clusters included in the calculation, are shown for acetic acid in Fig. S1 of the Supplement. For acetic acid, we found the best fit between experimental and estimated activity coefficients using $c_H = 0 \ \mathrm{kJ \ mol^{-1}}$ for both the water dimerization and acid hydration reactions. Decreasing the $c_H$ of either clustering reaction leads to stronger deviation from ideality, which in our case leads to a worse fit for water activity coefficient, and positive parameter values cannot be used to lower the interaction enthalpy. The

effective equilibrium constant for water dimer formation ($5.71 \times 10^5$) is below that of acetic acid hydration ($4.36 \times 10^6$), which explains why the fit parameter of the water dimer hydration should be higher (or equal, since positive values are not possible)
than the parameter for acid hydrate formation. Additionally, we calculated UNIFAC predictions of acid and water activity coefficients using AIOMFAC-web (AIOMFAC-web, 2020; Zuend et al., 2008, 2011). These calculations (without inorganic ions) correspond to modified UNIFAC calculations by Peng et al. (2001). From Fig. S1 we see that, for acetic acid, the UNIFAC model underestimates the experimental activity coefficients more than even the COSMO-RS estimate. Similar to COSMO-RS, UNIFAC is not able to predict the increasing trend of water activity coefficients with the increasing acid mole fraction.

For the other monocarboxylic acids studied here, we used the same $c_H$ value for water dimerization that was found for the acetic acid–water system, and fitted the $c_H$ of acid hydrate formation to the experimental activity coefficients of water and the acids in the corresponding acid–water systems (Hansen et al., 1955). The enthalpic parameter values of acid hydration used to estimate the activity coefficients shown in Fig. 4 are 0.0, 0.0, -10.5, -14.6, -9.2 and -8.4 kJ mol$^{-1}$, for formic ($n = 1$), acetic ($n = 2$), propanoic ($n = 3$), butanoic ($n = 4$), pentanoic ($n = 5$) and hexanoic ($n = 6$) acid, respectively. For formic acid, we
used the same $c_H$ parameter as for acetic acid due to lack of experimental activity coefficients. If the enthalpic parameters in COSMO-RS-DARE calculations are not fitted to experimental activity coefficients and instead are set to zero, the activity coefficients of both acid and water underestimate the experimental activity coefficients of Hansen et al. (1955) (see Fig. S2 of the Supplement). If no experimental activity coefficients are available for fitting the COSMO-RS-DARE parameters, COSMO-RS estimates agree with experiments overall better than COSMO-RS-DARE or UNIFAC. COSMO-RS-estimated acid activity
coefficients are close to the measured values in all mixing states, and for water activity coefficients the agreement between COSMO-RS and experiments is good in mixing states with $x_{\mathrm{acid}} < 0.75$.

Sachsenhauser et al. (2014) used the COSMO-RS-DARE method for binary systems containing either acetic ($n = 2$) or propanoic ($n = 3$) acid and a non-polar organic solvent. Their calculations show that the dimerization parameter (equivalent to $c_H$ in our calculations) is higher for propanoic acid than for acetic acid. This is opposite to what we observed for the hydration
parameters, where $c_H$ was found to be higher for acetic acid than for propanoic acid. This indicates that the fit parameters of one clustering reaction cannot be used to estimate the corresponding fit parameters of another clustering reactions of the same compound.

While COSMO-RS is fully predictive, COSMO-RS-DARE requires parameter fitting using experimental data. Fitted COSMO-RS-DARE parameters from one system can be used in other systems where the same clustering reactions are relevant. For
instance, Sachsenhauser et al. (2014) found that the same interaction parameters of acid dimers can be used in systems containing other similar (non-polar) solvents. This indicates that our interaction enthalpies can be applied to other aqueous systems, for instance, ternary systems containing an inorganic salt, in addition to the carboxylic acid and water. This would allow for extending the findings of this study to atmospherically relevant aerosol solutions.

The increasing length of the acid carbon backbone leads to larger deviation from ideality ($\gamma = 1$) for both the acid and water.
In convention I, this means the acid and water activity coefficient values are higher in mixtures containing the longer acids than the shorter acids. We observe that COSMO-RS-DARE estimated activity coefficients agree well with the experiments once the $c_H$ parameter is fitted. However, when the hydrate and water dimer reactions are included, COSMO-RS-DARE

is not able to predict realistic activity coefficients for water at high mole fractions ($x_{\mathrm{acid}} > 0.9$) of the acids. This is likely due to the low concentration of water in the binary solution, leading to errors in the description of the interactions between

water molecules. Still, COSMO-RS-DARE estimates agree well with the experiments at least up to 0.9 mole fraction of the monocarboxylic acids. This is an improvement compared to the UNIFAC model, which fails to reproduce experimental water activity coefficients already at acid mole fractions above 0.25. At very high acid mole fractions ($x_{\mathrm{acid}} > 0.95$), COSMO-RS-DARE predicts several orders of magnitude higher activity coefficients than what was seen in experiments.

### 3.2.2  Dicarboxylic acids

We tested the effect of including different clusters in the activity coefficient calculation of malonic acid ($m = 3$). A comparison between the experimental, UNIFAC-modeled and COSMO*therm*-estimated activity coefficients are shown in Fig. 5. The malonic acid activity coefficients are compared in convention III (Fig. 5a) and in convention I (Fig. 5b). In convention III, acid activity coefficients are given with respect to a $1 \ \mathrm{mol \ kg^{-1}}$ solution reference state (see Supplement for more information). The COSMO*therm*-estimated water activity coefficients are compared with experimental bulk (Fig. 5c) and particle (Fig. 5d)

phase activity coefficients and UNIFAC-estimated activity coefficients.

For malonic acid (and other studied dicarboxylic acids, see Figs S3-S5 of the Supplement), COSMO-RS-DARE is not able to improve the agreement between experiments and COSMO*therm* estimates, the best overall fit is found using COSMO-RS. The water activity coefficients estimated using COSMO-RS are close to ones estimated using the UNIFAC model (modified UNIFAC; Peng et al., 2001). Similarly to what has been seen with the UNIFAC model, COSMO-RS is able to predict water

activity coefficients obtained from bulk and evaporation (supersaturated) measurements.

Figure S4 of the Supplement shows comparisons between experimental and COSMO*therm*-estimated water activity coefficients in oxalic, adipic and pimelic acid. For these three acids, only water activities have been determined experimentally (Braban et al., 2003; Maffia and Meirelles, 2001; Marsh et al., 2017; Peng et al., 2001). In addition, water activities in adipic and pimelic acid solutions were only measured in bulk solutions (Marsh et al., 2017). We found a good agreement between

the bulk measurements and COSMO-RS-estimated water activity coefficients, with COSMO-RS slightly overestimating the experiments. This result is in line with previous comparisons of hydroxy carboxylic acids (Hyttinen and Prisle, 2020).

The COSMO-RS-estimated activity coefficients of the studied dicarboxylic acids are shown in Fig. 6. We can see that, using convention I, the activity coefficients of the smaller dicarboxylic acids are lower than of the larger dicarboxylic acids. Comparing COSMO-RS (solid lines) and UNIFAC estimates (dotted lines), there is less variation between the UNIFAC-

estimated activity coefficients for the different acids studied than between the COSMO-RS estimates. This indicates that, in COSMO-RS, the number of carbon atoms has a larger effect on activity coefficients than estimated by UNIFAC.

Additionally, we computed activity coefficients with consideration of the first dissociation step for oxalic acid (the most acidic dicarboxylic acid of this study) with dissociation of oxalic acid included in the COSMO-RS calculation. In this case, the system contains neutral oxalic acid ($H_2A$) and water ($H_2O$), as well as singly or doubly deprotonated oxalic acid ($HA^-$ or

$A^{2-}$, respectively) and hydronium ion ($H_3O^+$) according to the dissociation equilibrium

$$H_2A + 2H_2O \rightleftharpoons HA^- + H_3O^+ + H_2O \rightleftharpoons A^{2-} + 2H_3O^+ \tag{R2}$$

While both acid groups of oxalic acid can dissociate, here we consider only the first deprotonation, because the second dissociation constant of oxalic acid in water is higher (3.81; Rumble, 2018) than the first one (1.25; Rumble, 2018) and has a smaller effect on the equilibrium. Figure S6 of the Supplement shows the difference between activity coefficients in a system where

dissociation of oxalic acid is included and the binary system containing only neutral compounds. The calculation procedure is explained in more detail in the Supplement. There is no large difference in water activity coefficients when the ions are added to the system. A small change is seen in the acid activity coefficients, especially in the concentrated solutions where the estimated mole fraction of dissociated acid and hydronium ion is high. For the other carboxylic acids studied here, the effect of including dissociation is likely to be smaller than for oxalic acid, due to the lower mole fractions of ions present in solutions

of less acidic compounds.

## 3.3 Aqueous solubility

We estimated the aqueous solubility of the monocarboxylic acids ($n = 1$–6) using the COSMO-RS-DARE method. Since activity coefficients are used in the equilibrium conditions of the LLE calculations, we used the same $c_H$ parameters that were fitted in the activity coefficient calculations, to determine whether the same parameter value can be used in LLE calcula-

tions. As a comparison, we computed the same solubilities using COSMO-RS. Based on previous COSMO*therm* calculations, Michailoudi et al. (2020) found a good agreement with experimental aqueous solubilities of fatty acids with even number of carbon atoms ($n = 2$–12). A comparison between experimentally determined aqueous solubilities and the COSMO*therm* estimates of monocarboxylic acids are shown in Fig. 7.

We see that using COSMO-RS-DARE, COSMO*therm* is able to predict the miscibility of the smaller monocarboxylic acids

($n = 1$–4) but the experimental solubilities of pentanoic ($n = 5$) and hexanoic ($n = 6$) acids are overestimated to a greater degree than when using COSMO-RS. On the other hand, COSMO-RS underestimates the experimental solubility of butanoic acid by a factor of 18, while COSMO-RS-DARE overestimates the experimental solubilities (upper limit) of pentanoic and hexanoic acids only by factors of 3.4 and 4.1, respectively.

For dicarboxylic acids, we estimated aqueous solubilities using COSMO-RS. The COSMO*therm*-estimated and experimen-

tal solubilities are shown in Fig. 8. Different experimental heat of fusion and melting point values have been reported for some of the studied dicarboxylic acids. We calculated the lower and upper limit free energies of fusion by combining the different experimental values, and the aqueous solubilities were estimated using the two different free energy of fusion values. The higher $\Delta G_{fus}$ estimate gives a lower aqueous solubility. The variability in the COSMO*therm*-estimated solubilities is smaller than in the experimental solubilities.

The COSMO-RS solubility estimates of most of the dicarboxylic acids ($m = 3$–7) are within the range of experimentally determined solubilities. Using all lowest-energy conformers (up to 40 conformers), instead of only conformers containing no intramolecular hydrogen bonds, lowers the solubility estimates of all acids by a factor of 1.2 on average. The same effect

of including conformers containing intramolecular hydrogen bonds has been previously seen in aqueous solubilities of citric, tartaric, malic and maleic acid, and multifunctional organosulfates (Hyttinen and Prisle, 2020).


## 4 Conclusions

We compared COSMO*therm*-estimated activity coefficients and aqueous solubilities of simple carboxylic acids with experimental values and a commonly used UNIFAC model, and generally found a good agreement between experiments and COSMO-RS estimates. Using COSMO-RS-DARE, we were able to further improve the agreement between estimated and experimental water activity coefficient in binary monocarboxylic acid–water systems significantly compared to using COSMO-RS or UNIFAC. The COSMO-RS estimates of monocarboxylic acid activity coefficient in aqueous solutions agree with the experiments quite well, and were further improved by COSMO-RS-DARE when the enthalpic fitting parameters were fitted using experimental activity coefficients. We were also able to estimate activity coefficients of pentanoic and hexanoic acids using only experimental water activity coefficients in the fitting of the COSMO-RS-DARE enthalpic parameters. In addition, COSMO-RS-DARE was able to predict the miscibility of butanoic acid in water (using the fitting parameters of activity coefficient calculations), while COSMO-RS predicted a finite solubility. However, in aqueous solubility calculations of pentanoic and hexanoic acid, COSMO-RS led to a better agreement between the experiments and estimates compared to COSMO-RS-DARE.

For dicarboxylic acid–water systems, COSMO-RS produced better agreement with experiments than COSMO-RS-DARE. The experimental water activity coefficients from different sources have large variations and COSMO-RS-estimated water activity coefficients fit within the range of experimental water activity coefficients obtained from bulk and evaporation measurements. We also found a good agreement between COSMO-RS-estimated and experimental acid activity coefficients at all acid mole fractions.

COSMO-RS was able to reproduce the same even-odd behavior of the dicarboxylic acid properties that have previously been seen experimentally in vapor pressures (Bilde et al., 2003) and solubilities (Zhang et al., 2013), and computationally in gas-phase dimer formation (Elm et al., 2019). The calculated even-odd behavior observed here in aqueous solubilities is likely partially due to the even-odd variation of the melting points and heats of fusion. There is also no visible even-odd behavior in the COSMO-RS-estimated activity coefficients of the dicarboxylic acids. However, even-odd variation is seen in the effective equilibrium constants of dimerization of the larger dicarboxylic acids ($m \geq 4$), which do not rely on experimental properties.

Mono- and dicarboxylic acids are very common in the atmosphere and often used as model compounds for oxygenated functionalities in a range of applications from vapor pressure, condensation-evaporation, cloud condensation nuclei activity and hygroscopicity, but also aerosol phase and heterogeneous reactivity (Prenni et al., 2001; McNeill et al., 2008; Schwier et al., 2012; Rossignol et al., 2016). Solubilities and activity coefficients of these secondary organic aerosol (SOA) constituents are needed to accurately predict their activities, to determine central properties such as composition, phase state, and chemical reactivity. Accurate computational tools are critical to provide this information for systems where experimental data are not readily accessible in literature or by experimental design. We showed that COSMO*therm* provides a good solution to estimating thermodynamic properties of atmospherically relevant organic compounds that are not commercially available for measure-

ments. In addition to simple binary systems studied here, COSMO*therm* can be used to predict liquid-phase properties, such as activity coefficients, in complex, atmospherically relevant systems.

*Data availability.* The research data have been deposited in a reliable public data repository (the CERN Zenodo service) and can be accessed at https://doi.org/10.5281/zenodo.3842593 (Hyttinen et al., 2020).

375 *Author contributions.* NH performed the COSMO*therm* calculations. NH, RH and JE generated input files for the COSMO*therm* calculations. NH and NLP analyzed the results with contributions from all co-authors. NH wrote the first version of the manuscript and response to reviewers with contributions from all co-authors. NLP conceived, planned, supervised and secured funding for the project.

*Competing interests.* The authors declare that they have no conflict of interest.

*Acknowledgements.* We thank CSC - IT Center for Science, Finland, for computational resources.

380 *Financial support.* This project has received funding from the European Research Council (ERC) under the European Union's Horizon 2020 research and innovation programme, Project SURFACE (Grant Agreement No. 717022). Nønne L. Prisle, Noora Hyttinen, Reyhaneh Heshmatnezhad and Theo Kurtén also gratefully acknowledge the financial contribution from the Academy of Finland, including Grant No. 308238, 314175 and 315600. Jonas Elm is grateful for financial support from the Swedish Research Council Formas project number 2018-01745-COBACCA.

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

**Table 1.** List of the studied $\alpha,\omega$-dicarboxylic acids and their experimentally determined melting points and heats of fusion. The values were measured using anhydrous acids (crystalline at 298.15 K). The melting point of oxalic acid reported by (Booth et al., 2010) (370 K) is likely the temperature of the phase transition from dihydrate to anhydrous crystal polymorph. Similar transition was seen by (Omar and Ulrich, 2006) at 378.35 K in their differential scanning calorimetry experiment.

| Acid | Chemical formula | Melting point (K) | Heat of fusion (kJ mol$^{-1}$) |
|---|---|---|---|
| Oxalic | $C_2H_2O_4$ | 462.65[a]; 465.26[b] | 58.158[b] |
| Malonic | $C_3H_4O_4$ | 406[c]; 408.15[a] | 18.739[c] |
| Succinic | $C_4H_6O_4$ | 455.2[d]; 458[c]; 461.15[a] | 31.259[c]; 34.0[d] |
| Glutaric | $C_5H_8O_4$ | 363.9[d]; 369[c]; 372.15[a] | 18.8[d]; 22.043[c] |
| Adipic | $C_6H_{10}O_4$ | 419.0[d]; 423[c]; 426.15[a] | 33.7[d]; 35.891[c] |
| Pimelic | $C_7H_{12}O_4$ | 368.2[d]; 379.15[a] | 23.7[d] |
| Suberic | $C_8H_{14}O_4$ | 413.2[d]; 417.15[a] | 30.7[d] |

[a]Cornils and Lappe (2000); [b]Omar and Ulrich (2006); [c]Booth et al. (2010); [d]Roux et al. (2005)

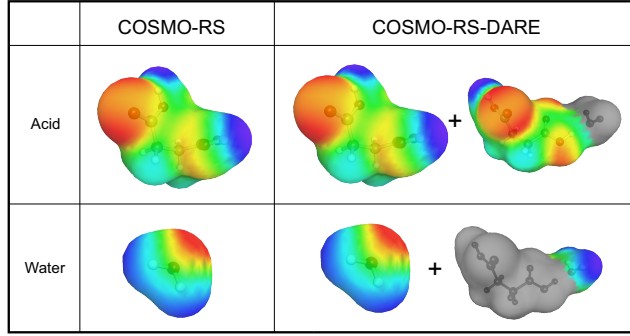

**Figure 1.** The $\sigma$-surfaces of succinic acid and water conformers used in COSMO-RS and COSMO-RS-DARE calculations. The conformer distributions in COSMO-RS-DARE include parts of cluster $\sigma$-surfaces (in this example a hydrate cluster). Color coding of $\sigma$-surfaces: red = negative partial charge, blue = positive partial charge, green = neutral partial charge, grey = omitted $\sigma$-surface.

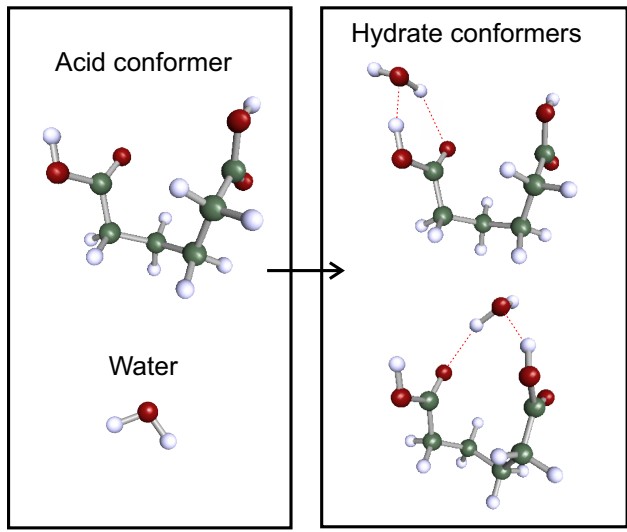

**Figure 2.** The formation of dicarboxylic acid hydrate conformers. Color coding: green = C, white = H, red = O.

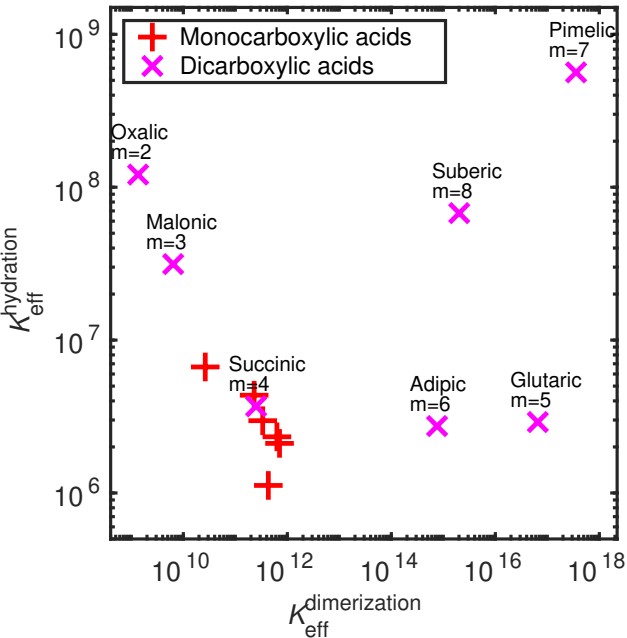

**Figure 3.** Dimensionless effective equilibrium constants of acid dimer and hydrate formation ($K_{\text{eff}}^{\text{dimerization}}$ and $K_{\text{eff}}^{\text{hydration}}$, respectively) in condensed phase, at 298.15 K. See Table 2 for the values.

**Table 2.** Dimensionless effective equilibrium constants ($K_{\text{eff}}$) of cluster formation in condensed phase, at 298.15 K.

| Acid | Carbon # | $K_{\text{eff}}^{\text{hydration}}$ | $K_{\text{eff}}^{\text{dimerization}}$ |
|---|---|---|---|
| Formic | 1 | $6.67 \times 10^6$ | $2.64 \times 10^{10}$ |
| Acetic | 2 | $4.36 \times 10^6$ | $2.30 \times 10^{11}$ |
| Propanoic | 3 | $1.12 \times 10^6$ | $4.31 \times 10^{11}$ |
| Butanoic | 4 | $2.97 \times 10^6$ | $3.36 \times 10^{11}$ |
| Pentanoic | 5 | $2.11 \times 10^6$ | $7.08 \times 10^{11}$ |
| Hexanoic | 6 | $2.33 \times 10^6$ | $6.30 \times 10^{11}$ |
| Oxalic | 2 | $1.21 \times 10^8$ | $1.35 \times 10^9$ |
| Malonic | 3 | $3.14 \times 10^7$ | $6.38 \times 10^9$ |
| Succinic | 4 | $3.69 \times 10^6$ | $2.51 \times 10^{11}$ |
| Glutaric | 5 | $2.91 \times 10^6$ | $6.56 \times 10^{16}$ |
| Adipic | 6 | $2.74 \times 10^6$ | $7.58 \times 10^{14}$ |
| Pimelic | 7 | $5.64 \times 10^8$ | $3.57 \times 10^{17}$ |
| Suberic | 8 | $6.77 \times 10^7$ | $2.02 \times 10^{15}$ |
| Water | - | - | $5.71 \times 10^5$ |

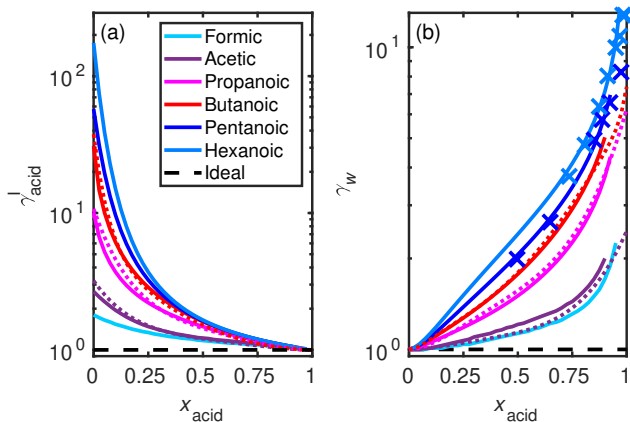

**Figure 4.** Activity coefficients ($\gamma^{\mathrm{I}}$) of (a) monocarboxylic acids and (b) water in all mixing states of the binary aqueous solutions, at 298.15 K. The solid lines represent activity coefficient estimates using COSMO-RS-DARE, dotted lines are calculated from the equations fitted to experiments by Hansen et al. (1955), and the markers are the experimental points from the same study. For the studied acids with finite aqueous solubilities at 298.15 K (pentanoic and hexanoic acid), water activity coefficients were measured using acid-rich solutions ((Hansen et al., 1955)). The water activity coefficients at high $x_{\mathrm{acid}}$ are not shown in the figure, because COSMO-RS-DARE overpredicts the experiments by several orders of magnitude. All activity coefficient values are given in Tables S3 and S4 of the Supplement.

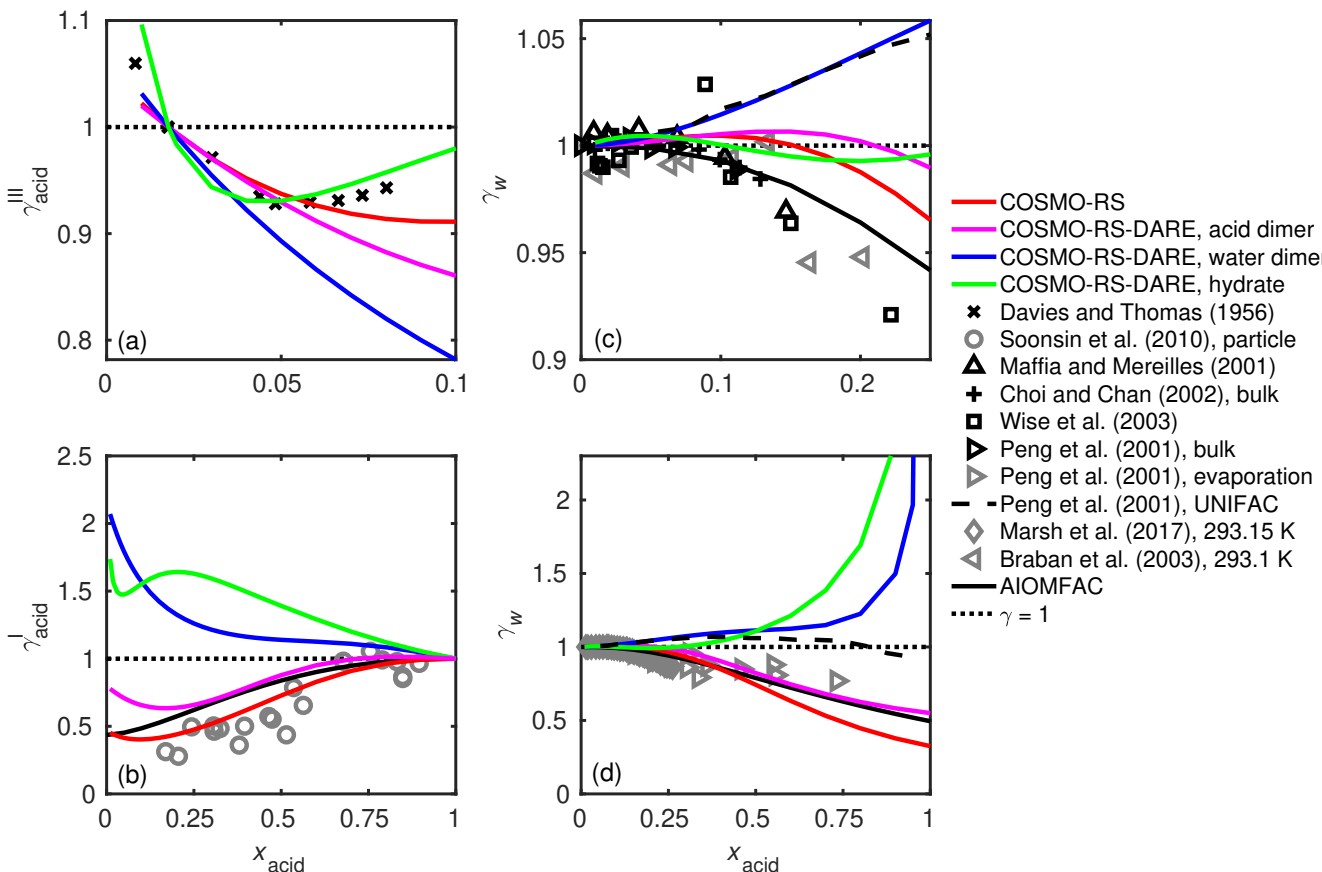

**Figure 5.** Activity coefficients of (a-b) malonic acid and (c-d) water in the binary mixtures at 298.15 K calculated using different clustering reactions in the COSMO-RS-DARE calculation. As a comparison are activity coefficients of malonic acid by Davies and Thomas (1956) (at 298.15 K given in convention III) and Soonsin et al. (2010) (particle measurements at various temperatures given in convention I) and of water by Maffia and Meirelles (2001), Choi and Chan (2002), Wise et al. (2003), Peng et al. (2001), Marsh et al. (2017), Braban et al. (2003) and AIOMFAC-web (2020).

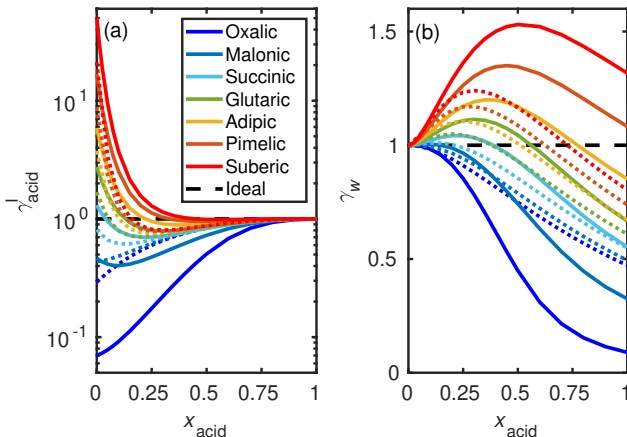

**Figure 6.** COSMO-RS- (solid lines) and UNIFAC-estimated (dotted lines; AIOMFAC-web, 2020) activity coefficients ($\gamma^I$) of (a) dicarboxylic acids and (b) water in the binary acid–water mixtures at 298.15 K. All COSMO*therm*-estimated activity coefficient values are given in Tables S5 and S6 of the Supplement.

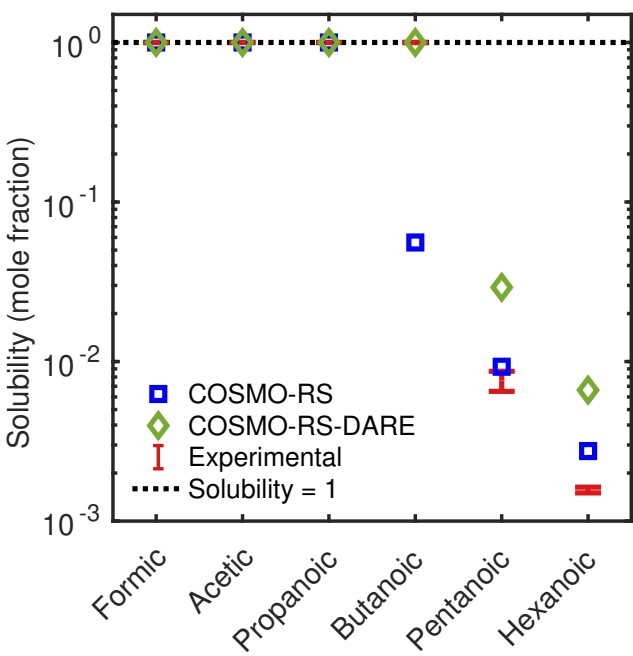

**Figure 7.** COSMO*therm*-estimated aqueous solubility of monocarboxylic acids calculated using COSMO-RS and COSMO-RS-DARE, at 298.15 K. Experimental solubilities by Saxena and Hildemann (1996) ($n = 1$–6) and Romero and Suárez (2009) ($n = 5$–6).

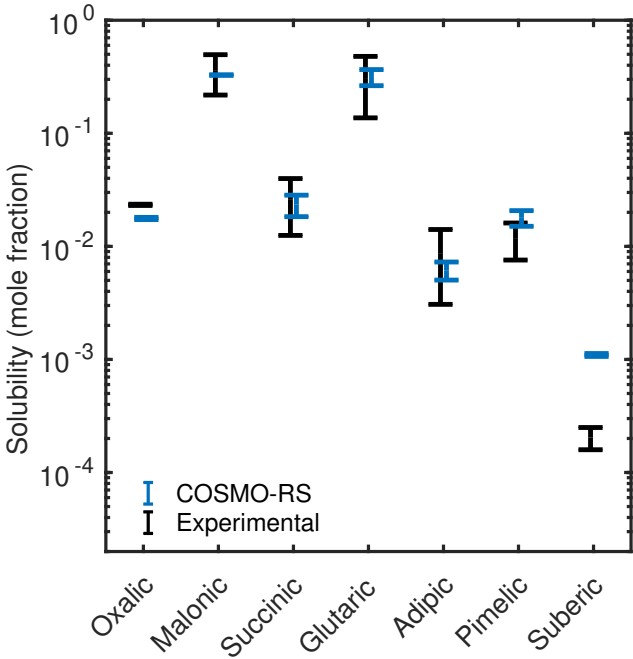

**Figure 8.** Estimated solubilities of the dicarboxylic acids compared to their experimental solubilities by Apelblat and Manzurola (1987) ($m$ = 2–4, 6), Apelblat and Manzurola (1989) ($m$ = 5), Apelblat and Manzurola (1990) ($m$ = 7–8), Saxena and Hildemann (1996) ($m$ = 2–6), Omar and Ulrich (2006) ($m$ = 2), Zhang et al. (2013) ($m$ = 2–8), Brooks et al. (2002) ($m$ = 4), Song et al. (2012) ($m$ = 5) and O'Neil (2013) ($m$ = 5), at 298.15 K. The lower and upper limit solubility estimates are obtained using the highest and lowest free energies of fusion (estimated from experimental melting point and heat of fusion in Table 1), respectively.