# Peer review of "Technical note: Estimating aqueous solubilities and activity coefficients of mono- and $\alpha,\omega$ -dicarboxylic acids using COSMO*therm"

_Atmospheric Chemistry and Physics, 2020_

## Referee Comment (RC1) · Anonymous Referee #1 · 10 Jun 2020

This Technical Note by Hyttinen et al. introduces a new approach for the simulation of activity coefficients and aqueous solubilities of mono- and dicarboxylic acids using the COSMOtherm software. The COSMO-RS-DARE model extension is used in this work, which includes hydration and dimerization of the organic acids in water.

The paper is rather technical and focused on the choices made in this application of COSMO-RS-DARE. This is appropriate for a technical note. However, I found it difficult to judge the added value of this work in terms of the broader application of such methods in an entirely predictive way, e.g. to generate activity coefficient data sets for more complex mixtures of organic acids that have not been studied experimentally.

[Figure]

The manuscript does not offer much advise in this respect nor a discussion about the applications of the model. Adding a discussion about strengths and weaknesses of the introduced method would improve the paper.

The writing and structure of this technical note are overall appropriate and the topic is of interest for the ACP community. However, as indicated by several of my general and specific comments below, there are a number of mistakes, inconsistencies and unclear statements that need to be addressed before publication can be recommended.

**1  General comments**

- Aerosol acidity is mentioned in the introduction, but beyond that I could not find any description of the approach in COSMO-RS-DARE or COSMO-RS to account for partial dissociation of organic acids in solution. It would seem to be important as an effect that may compete with dimerization and hydration of undissociated acids. Please discuss.

- Fitting of reaction enthalpy parameters to existing experimental data was carried out, making the method perhaps less predictive than one would hope for. It is unclear how important the fitting of COSMO-RS-DARE model parameters is to achieve the presented activity coefficient and water-solubility results. If such fit parameters are essential, could you discuss the advantages of the COSMOtherm modelling approach compared to more traditional fitting of activity coefficient models for binary solutions, such as a Van Laar model or group-contribution models like UNIFAC?

- The provided quantitative comparison to available measurements of dicarboxylic acid activity coefficients or water activities is rather limited and many existing data for concentrated aqueous solutions, e.g., by Choi and Chan, J. Phys. Chem. A

2002, 106, 4566–4572 or Marsh et al. (2017), could be used for a direct comparison of measured and predicted water activities / activity coefficients of the studied binary solutions. In addition, a comparison to predictions from other models, such as the UNIFAC / AIOMFAC / E-AIM models or other such approaches would allow the reader to compare the performance of COSMO-RS-DARE to such parameterized thermodynamic models that are often used in this community.

**1.1 Specific comments**

- Abstract: The abstract would benefit from a quantitative statement about the average accuracy of the COSMO-RS-DARE predictions of activities or activity coefficients compared to the available experimental data and/or predictions by the standard COSMOtherm / COSMO-RS model. It would also be useful to state whether the outlined COSMO-RS-DARE method is fully predictive or not.

- Line 31: Clarify the sentence with "the Gibbs-Duhem equation was fitted"; what exactly was fitted? To my knowledge, the fundamental Gibbs-Duhem equation has no dedicated fit parameters.

- L. 32 – 34: The sentence should be improved given that, among the cited references, both Peng et al. (2001) and Choi and Chan (2002) use electrodynamic balance measurements that cover mass fractions of the solute far beyond the dilute solution range of the dicarboxylic acids. Further, the COSMOtherm predictions provided in this work (Fig. 3) seem not to be compared to such experimental data sets, even though the authors are aware of them.

- L. 35: Define abbreviations for COSMOtherm and COSMO-RS.

- L. 70: Clarification necessary; Eq. (1), (2) define activity coefficients using the pseudo-chemical potential, which differs from the "regular" chemical potential

more often used. However, comparison of Eqs. (1) and (2) raises the question how $\mu_i^{*\circ,I}$ in Eq. (1) differs from $\mu_i^\circ$ in Eq. (2)? Eq. (2) seems to express the same relationship as Eq. (1). How exactly do they differ and what convention is used for the activity coefficients in Eq. (2)? Also, on line 71, the gas constant is expressed using kcal for energy; use of SI units would generally be preferred.

- L. 76 – 79: Clarify the basis for Eq. (3), why should that apply (references / reasons)? Also, Eq. (4) stated in the current form seems incorrect and awkward: why write the right hand side composition as $a_i^{I,\beta}(1 - x_{SOL,w})$? This seems not entirely correct and to be a potential source of confusion. This confusion exists because $(1 - x_{SOL,w})$ should be exactly the same as $x_{SOL,acid}$ in a binary mixture (in the same phase), yet the former expression would only be correct for binary aqueous mixtures, not in general. However, in the LLE case, the mole fractions of acid in phases $\alpha$ and $\beta$ will differ, which is missing in Eq. (4). Why not write $a_i^{I,\beta}(x_{SOL,acid}^\beta)$ and analogously for phase $\alpha$. Further, please define the meaning of subscript SOL.

- L. 98: The following statement "by multiplying the reaction equilibrium constant with the ratio of the activity coefficients..." and need for Eq. (11) seem unwarranted and require further explanation. Why should Eq. (11) be necessary? Is this because the authors are only considering mole fractions in the reaction constant, not activities? It is unclear because based on Eq. (8) – (10), in which chemical potentials and therefore implied activity coefficients are used, there seem to be no need for Eq. (11). Are you instead using pseudo-chemical potentials? Please clarify this. Also, activity coefficients of the reactants and products in what mixture? If the mixture contains a solvent, e.g. water, which affects the values of activity coefficients, how would Eq. (11) become solvent-independent?

- L. 114: "equilibrium constants on a mole fraction basis" – Unclear what "mole fraction basis" should imply here. If the equilibrium constant is computed based

on free energies of reaction or chemical potentials of reaction, they are always dimensionless – as any thermodynamic equilibrium constant should always be (concentration-product-based equilibrium constants are only approximations and not thermodynamically correct).

- L. 130 – 133: Eq. (15) and text: It is unclear why enthalpic and entropic energy contributions should not already be accounted for by $\Delta^0$. Also why is there a factor 2? It seems possible that the energy difference is just not known/predicted well enough, such that a fit parameter was introduced to match experimental data. Is that the motivation for the "enthalpic" contribution in Eq. (15)? Please discuss.

- L. 180 – 184: The discussion of hydration and dimerization in aqueous solutions raises the question whether the dissociation of the carboxylic acids into dissolved ions was considered in the simulation? In dilute aqueous solution, dissociation of the acid and formation of hydronium ions would seem to be favorable over non-dissociative hydration. Please discuss. Acidity and pH are mentioned in the introduction, but nothing is said about acid dissociation within COSMO-RS-DARE simulations.

- L. 198 – 199: Statement: "We used these experimental activity coefficients to fit the enthalpic parameters ($c_H$) for each of the acids in the COSMO-RS-DARE calculations." Given that such fit parameters were introduced, how predictive is the outlined COSMO-RS-DARE method for calculation of activity coefficients in aqueous solutions for compounds where the experimental equilibrium constants or activity coefficients are not known? Elaborating the discussion on this key point seems important for applications in atmospheric aerosol modeling. For example, if these parameters were all ignored (set to zero), how different would the predictions be for activity coefficients and solubilities in water?

- L. 214, 215: Do these calculation include phase separation for the larger acids

or is a mixed phase assumed for all compositions? At higher organic acid mole fractions, the binary mixtures show high values for water activity coefficients, as also stated in the caption to Fig. 2. This may suggest liquid-liquid phase separation could occur, which could affect the interpretation of the experimental data used for comparison. Please discuss.

- L. 261: "However, without experimental activity coefficient data of oxalic acid–water systems, we are not able to fit the $c_H$ parameter needed for reliable estimates." – This statement is simply not correct. A literature search reveals that there exist multiple useful measurements for this binary system, such as water activities at dilute and concentrated conditions, from several sources. These include the work of Maffia and Meirelles (2001) mentioned earlier in the study, as well as work by Braban et al. (2003) and Marsh et al. (2017), which provide water activity data that should be used here.

- L. 264 and Fig. 3: The partial dissociation of dicarboxylic acids is either not considered at low acid concentrations or not discussed, even though dissociation would seem to be likely, especially for the smaller acid molecules. Was it determined to be irrelevant? This will require adequate discussion. Also, mention that mole fractions used in this work are defined assuming undissociated acid molecules (if this is indeed the case).

- L. 290 – 294: On melting temperatures and solids considered:
"Cornils and Lappe (2000) and Omar and Ulrich (2006) also measured the melting point of oxalic acid and found temperatures almost 100 K higher than Booth et al. (2010). ..." In this context, are the differences in melting points due to unclear statements about the crystalline form of oxalic acid (anhydrous or dihydrate)? Omar and Ulrich (2006) show in their paper that the solid–solid phase transition from the dihydrate to the anhydrous oxalic acid crystal polymorph occurs around 378 K, which is close to the melting temperature stated by Booth et al., while

the anhydrous oxalic acid melts at 465 K. Therefore, the correct values of use in the COSMOtherm simulations will depend on the temperature range of interest; for room temperature, using the dihydrate from is very likely the crystalline solid to be considered and consequently the solubility equilibrium should be solved for oxalic acid dihydrate not the anhydrous form. It is also clear that the actual melting temperature and enthalpy of fusion is not as uncertain as the current text implies. Using the correct equilibrium relations for the dihydrate the agreement between COSMOtherm and the measured solubility data would be expected to be much better than shown in Fig. 3.

- L. 305: "to significantly improve the activity coefficient estimates ..." – compared to what? COSMO-RS or other methods? What about a comparison to methods like UNIFAC, such as the model by Peng et al. (2001).

- L. 308: "In addition, COSMO-RS-DARE is able to predict the miscibility of butanoic acid in water, while COSMO-RS predicts a finite solubility" – This is nice. However, how much of that success comes from fitting model parameters ($c_H$) rather than the use of the DARE extension?

- Table 1: State whether the listed data are for the pure anhydrous solids of the acids or for hydrates (especially in case of oxalic acid). Also replace "literature values" by more appropriate wording.

- Table 2: state the temperature for the listed equilibrium constants and for completeness also for which phase / solvent they apply (given that there is also gas phase dimerization of such acids).

- Figure 3: Why are the prediction data cut of at the solubility limits? It would seem useful to mark the solubility limit at 298.15 K for each acid, but to also show the predictions for the supersaturated range (which may apply in aerosols). This would also facilitate an extended comparison with experimental data existing for

those higher concentrations, e.g. by Choi and Chan (2002) and Marsh et al. (2017). With the chosen log-scale for the x-axis, too much emphasis is put on the very dilute concentration range below $10^{-3}\ x_{acid}$, which seems not to be insightful.

**2 References mentioned**

Braban, C. F., Carroll, M. F., Styler, S. A., and Abbatt, J. P. D.: Phase Transitions of Malonic and Oxalic Acid Aerosols, The Journal of Physical Chemistry A, 107, 6594-6602, 10.1021/jp034483f, 2003.

Choi, M. Y., and Chan, C. K.: Continuous Measurements of the Water Activities of Aqueous Droplets of Water-Soluble Organic Compounds, The Journal of Physical Chemistry A, 106, 4566-4572, 10.1021/jp013875o, 2002.

Maffia, M. C., and Meirelles, A. J. A.: Water Activity and pH in Aqueous Poly-carboxylic Acid Systems, Journal of Chemical & Engineering Data, 46, 582-587, 10.1021/je0002890, 2001.

Marsh, A., Miles, R. E. H., Rovelli, G., Cowling, A. G., Nandy, L., Dutcher, C. S., and Reid, J. P.: Influence of organic compound functionality on aerosol hygroscopicity: dicarboxylic acids, alkyl-substituents, sugars and amino acids, Atmos. Chem. Phys., 17, 5583-5599, 10.5194/acp-17-5583-2017, 2017.

---

## Referee Comment (RC2) · Anonymous Referee #2 · 26 Jun 2020

General comment

This paper employs the recently developed COSMO-RS-DARE model to estimate activity coefficients and solubilities of carboxylic acids in water. COSMO-RS-DARE is an extension of COSMOtherm that takes dimerization and aggregation in solution explicitly into account. This technical note concludes that COSMO-RS-DARE leads to better agreement with experimental data than COSMOtherm for the investigated mixtures.

Although this paper is submitted as a technical note, the technical description of COSMOtherm and its extension COSMO-RS-DARE is lacking a proper derivation and explanation. Also, the benefit of COSMO-RS-DARE compared with COSMOtherm re-

mains unclear. As it seems, the new method relies on experimentally determined activity coefficients to calculate dimerization equilibria. Therefore, the benefit of COSMO-RS-DARE in the absence of experimental data is unclear. It is not clear whether COSMO-RS-DARE just performs better in predicting solubilities because of an additional degree of freedom introduced through potential dimerizations or a more accurate description of the system.

Major revisions of the manuscript are required before this technical note can be considered for publication. The different COSMO versions need to be explained better and the discussion of the results needs to be improved.

Specific comments

Lines 20 – 22: Here, acidity is mentioned as highly relevant. But the approach used in this technical note totally neglects deprotonation of acids.

Line 25: activity data of carboxylic acid-water systems is abundant as exemplified by the studies mentioned just below this sentence and there are even more. Please revise this sentence.

Lines 65 – 66: the meaning of a pseudo-chemical potential should be explained.

Line 79: activity should be replaced by the activity coefficient in this equation.

Line 81: This equation should be derived or a reference should be given.

Line 96: How is the dielectric energy calculated or defined?

Line 96: The difference between the chemical potential and the pseudo chemical potential is not clearly made and not explained. Here, the same symbol is used to refer to the chemical potential that was used before for the pseudo chemical potential.

Line 100: Equation (11) needs to be explained better.

Line 112: The derivation of Eq. (14) remains obscure. The equation rather seems to

be a definition of the effective equilibrium constant than a derived equation.

Line 116: How is the surface of a molecule defined? Either explain here or give a reference.

Line 125: what is a property calculation?

Line 125: what is the screening charge density? A scheme might help to explain it.

Line 128 – 129: Why are interaction sites of molecule B not treated the same way?

Line 133 – 134: Why is the entropic parameter kept zero? This seems arbitrary. Please justify.

Lines 150 – 158: This section is difficult to understand. A scheme might help.

Lines 225 – 227: This finding questions the benefit of the method.

Lines 234: I would not refer to dicarboxylic acids as being of low aqueous solubility. Some dicarboxylic acids have a high solubility. Moreover, data well into the supersaturated range is available (e.g. in Soonsin et al., 2010). This sentence needs to be revised accordingly.

Line 247: Figs S2 and S3 should be moved to the main manuscript.

Line 255: Fig. S4 should be moved to the main manuscript.

Line 264: The logarithmic plot is not very informative. Rather show the figures from the SI here.

---

## Author Comment (AC1) · 2 Sep 2020

We thank the referees for their thoughtful and constructive comments which we believe have helped improve the manuscript substantially. We have revised the manuscript following the referees' suggestions. You can find answers to the referee comments (*in italics*) below with additions to the manuscript and supplement text (**in bold**). The added references are listed at the end. The line and figure numbers refer to the original manuscript.

**Anonymous Referee #1**

This Technical Note by Hyttinen et al. introduces a new approach for the simulation of activity coefficients and aqueous solubilities of mono- and dicarboxylic acids using the COSMOtherm software. The COSMO-RS-DARE model extension is used in this work,which includes hydration and dimerization of the organic acids in water.
The paper is rather technical and focused on the choices made in this application of COSMO-RS-DARE. This is appropriate for a technical note. However, I found it difficult to judge the added value of this work in terms of the broader application of such methods in an entirely predictive way, e.g. to generate activity coefficient data sets for more complex mixtures of organic acids that have not been studied experimentally. The manuscript does not offer much advise in this respect nor a discussion about the applications of the model. Adding a discussion about strengths and weaknesses of the introduced method would improve the paper. The writing and structure of this technical note are overall appropriate and the topic is of interest for the ACP community. However, as indicated by several of my general and specific comments below, there are a number of mistakes, inconsistencies and unclear statements that need to be addressed before publication can be recommended.

*Author's response: We have added discussion of the strengths and weaknesses, as well as suggestions for the applicability of COSMO-RS-DARE and COSMO-RS, as described in more detail below. We also changed the title from "Technical note: Estimating aqueous solubilities and activity coefficients of mono- and α,ω-dicarboxylic acids using COSMO-RS-DARE" to "Technical note: Estimating aqueous solubilities and activity coefficients of mono- and α,ω-dicarboxylic acids using COSMO-RS" in order to not emphasize the COSMO-RS-DARE method too much because majority of the results shown here are calculated using COSMO-RS.*

**Changes in manuscript (section 1, line 55): Most atmospherically relevant multifunctional compounds are not readily available for experimental determination of thermodynamic properties. Accurate theoretical estimates are therefore essential for advancing current aerosol process modeling to include more complex compounds and mixtures. Here, we demonstrate the applicability of COSMO-RS theory in calculating condensed-phase properties of atmospherically relevant organic compounds. Carboxylic acids are among the most abundant and well characterized organic compounds in the troposphere and are therefore a good compound class to use to validate the use of COSMO-RS in atmospheric research.**

**(section 4, line 330): We showed that COSMO*therm* provides a good solution to estimating thermodynamic properties of atmospherically relevant organic compounds that are not commercially available for measurements. In addition to simple binary systems studied here, COSMO*therm* can be used to predict liquid-phase properties, such as activity coefficients, in complex, atmospherically relevant systems.**

**1 General comments**

• Aerosol acidity is mentioned in the introduction, but beyond that I could not find any description of the approach in COSMO-RS-DARE or COSMO-RS to account for partial dissociation of organic acids in solution. It would seem to be important as an effect that may compete with dimerization and hydration of undissociated acids. Please discuss.

*Author's response: We agree with the reviewer that acid dissociation could be an important effect and its importance should be better described in the manuscript. A section about accounting for dissociation in calculation of oxalic acid and water activity coefficients was added.*

**Changes in manuscript (end of section 3.2.2, line 269): We additionally calculated activity coefficients of oxalic acid (the most acidic dicarboxylic acid of this study) with dissociation of oxalic acid included in the system. In this case, the system contains neutral oxalic acid (HA) and water ($H_2O$), as well as deprotonated oxalic acid ($A^-$) and hydronium ion ($H_3O^+$) according to the dissociation equilibrium**

$$HA + H_2O \leftrightarrow A^- + H_3O^+$$

**Figure S6 of the Supplement shows the difference between activity coefficients in a system where dissociation of oxalic acid is included and the binary system containing only neutral compounds. The calculation procedure is explained in more detail in the Supplement. There is no large difference in water activity coefficients when the ions are added to the system. A small change is seen in the acid activity coefficients, especially in the concentrated solutions where the estimated mole fraction of dissociated acid and hydronium ion is high. For the other carboxylic acids studied here, the effect of including dissociation is likely to be smaller than for oxalic acid, due to the lower mole fractions of ions present in solutions of less acidic compounds.**

**Changes in Supplement:**

[Figure]

**Figure S6: Activity coefficients of (a) oxalic acid and (b) water at 298.15 K calculated for systems where dissociation of oxalic acid was (red) and was not (blue) accounted for. On the x-axis, the mole fraction is for the total oxalic acid, meaning $x_{acid}^{DC}$ in the systems where dissociation is included and $x_{acid}$ in the system where dissociation is not taken into account. In both systems, the reference state for oxalic acid is the pure neutral acid and for water pure neutral water.**

• Fitting of reaction enthalpy parameters to existing experimental data was carried out, making the method perhaps less predictive than one would hope for. It is unclear how important the fitting of COSMO-RS-DARE model parameters is to achieve the presented activity coefficient and water-solubility results. If such fit parameters are essential, could you discuss the advantages of the COSMOtherm modelling approach compared to more traditional fitting of activity coefficient models for binary solutions, such as a Van Laar model or group-contribution models like UNIFAC?

*Author's response: We appreciate the comment and agree with the reviewer that the advantages of COSMOtherm modeling compared to group contribution methods should be included. We have added UNIFAC model estimates to the comparison between COSMOtherm-estimated and experimental activity coefficients.*

**Changes in manuscript (section 1, line 35): Group contribution methods, such as UNIFAC (Fredenslund et al., 1975) and AIOMFAC (Zuend et al., 2008), are often used to estimate activity coefficients of atmospherically relevant compounds.**
**(section 3.2.1, line 210): We additionally calculated UNIFAC predictions of acid and water activity coefficients using AIOMFAC-web (AIOMFAC-web, 2020; Zuend et al., 2008, 2011). These calculations (without inorganic ions) correspond to modified UNIFAC calculations by Peng et al. (2001). From Fig. S1 we see that, for acetic acid, the UNIFAC model underestimates the experimental activity coefficients more than even the COSMO-RS estimate.**

**Similar to COSMO-RS, UNIFAC is not able to predict the increasing trend of water activity coefficients with the increasing acid mole fraction.**
**(section 3.2.2, line 232): The water activity coefficients estimated using COSMO-RS are close to ones estimated using the UNIFAC model (modified UNIFAC; Peng et al. 2001). Similarly to what has been seen with the UNIFAC model, COSMO*therm* is able to predict water activity coefficients at low acid mole fractions, while at high acid mole fractions both models fail to reproduce experimental activity coefficients. This indicates that COSMO*therm* is not able to describe the water-acid interactions in supersaturated (crystalline) carboxylic acid mixtures.**
**(section 3.2.2, line 265): Comparing COSMO*therm* (solid lines) and UNIFAC estimates (dotted lines), there is less variation between the UNIFAC-estimated activity coefficients for the different acids studied than between the COSMO*therm* estimates. This indicates that, in COSMO*therm*, the number of carbon atoms has a larger effect on activity coefficients than estimated by UNIFAC.**

• The provided quantitative comparison to available measurements of dicarboxylic acid activity coefficients or water activities is rather limited and many existing data for concentrated aqueous solutions, e.g., by Choi and Chan, J. Phys. Chem. A 2002, 106, 4566–4572 or Marsh et al. (2017), could be used for a direct comparison of measured and predicted water activities / activity coefficients of the studied binary solutions. In addition, a comparison to predictions from other models, such as the UNIFAC / AIOMFAC / E-AIM models or other such approaches would allow the reader to compare the performance of COSMO-RS-DARE to such parameterized thermodynamic models that are often used in this community.

*Author's response: The missing experimental activity coefficients were added, in addition to UNIFAC and modified UNIFAC model estimations (Peng et al. 2001). Those are shown in Figs 5, 6, S2, S3 and S4. In addition, more discussion about the comparison between the different models and experiments was added.*

**Chances in manuscript (section 1, line 32): Activity coefficients of malonic, succinic and glutaric acid (m= 3, 4 and 5) have been measured by Davies and Thomas (1956) and Soonsin et al. (2010) in bulk and particle experiments, respectively.**
**(beginning of section 3.2.2, line 232): We tested the effect of including different clusters in the activity coefficient calculation of malonic acid (m = 3). A comparison between the experimental, UNIFAC-modeled and COSMO*therm*-estimated activity coefficients are shown in Fig. 5. The malonic acid activity coefficients are compared in convention III (Fig. 5a) and in convention I (Fig. 5b). The COSMO*therm*-estimated water activity coefficients are compared with experimental bulk (Fig. 5c) and particle (Fig. 5d) phase activity coefficients and UNIFAC-estimated activity coefficients. For malonic acid (and other studied dicarboxylic acids, see Figs S3-S5 of the Supplement), COSMO-RS-DARE is not able to improve the agreement between experiments and COSMO*therm* estimates, the best overall fit is found using COSMO-RS.**
**(section 3.2.2, line 232): Figure S4 of the Supplement shows comparisons between experimental and COSMO*therm*-estimated water activity coefficients in oxalic, adipic and pimelic acid. For these three acids, only water activities have been determined experimentally (Braban et al., 2003; Maffia and Mereilles, 2001, Marsh et al., (2017); Peng et al., 2001). In addition, water activities in adipic and pimelic acid solutions were only**

measured in bulk solutions (Marsh et al., 2017). We found a good agreement between the bulk measurements and COSMO-RS-estimated water activity coefficients, with COSMO*therm* slightly overestimating the experiments. This result is in line with previous comparisons of hydroxy carboxylic acids (Hyttinen and Prisle, 2020).

**Changes in Supplement:**

[Figure]

**Figure S3: Activity coefficients of (a-b) glutaric acid and (c-d) water in the binary mixtures at 298.15 K calculated using different clustering reactions in COSMO*therm* calculation. As a comparison are experimentally determined activity coefficients of malonic acid by Davies and Thomas[S4] (at 298.15 K given in convention III) and Soonsin et al.[S5] (particle measurements at various temperatures given in convention I) and of water by Peng et al.,[S6] Choi and Chan,[S7] Marsh et al.,[S8] Wise et al.[S9] and AIOMFAC-web.[S3]**

[Figure]

**Figure S4: Activity coefficients of (a-b) succinic acid and (c-d) water in binary mixtures at 298.15 K calculated using different clustering reactions in COSMO*therm* calculations. As a comparison are experimentally determined activity coefficients of malonic acid by Davies and Thomas[S4] (at 298.15 K given in convention III) and Soonsin et al.[S5] (particle measurements at various temperatures given in convention I) and of water by Peng et al.,[S6] Marsh et al.,[S8] Wise et al.,[S9] Maffia and Mereilles[S10] and AIOMFAC-web.[S3]**

1.1 Specific comments
• Abstract: The abstract would benefit from a quantitative statement about the average accuracy of the COSMO-RS-DARE predictions of activities or activity coefficients compared to the available experimental data and/or predictions by the standard COSMOtherm / COSMO-RS model. It would also be useful to state whether the outlined COSMO-RS-DARE method is fully predictive or not.

*Author's response: We agree that it would be beneficial to explicitly state whether the COSMO-RS-DARE method is fully predictive. The following has been added to the abstract:*

**Changes in manuscript (abstract, line 3): Conductor-like Screening Model for Real Solvents (COSMO-RS) underestimates experimental monocarboxylic acid activity coefficients by less than a factor of 2 but experimental water activity coefficients are underestimated more especially at high acid mole fractions.**
**(abstract, line 5): COSMO-RS-DARE is not fully predictive, but fitting parameters found here can be used to estimate thermodynamic properties of monocarboxylic acids in other aqueous solvents, such as salt solutions. For the dicarboxylic acids, COSMO-RS is sufficient for predicting aqueous solubility and activity coefficients and no fitting to experimental values is needed. This is highly beneficial for applications to atmospheric systems, as this data is typically not available for a wide range of mixing states realized in the atmosphere, either due to feasibility of the experiments or to sample availability.**

• Line 31: Clarify the sentence with "the Gibbs-Duhem equation was fitted"; what exactly was fitted? To my knowledge, the fundamental Gibbs-Duhem equation has no dedicated fit parameters.

*Author's response: We apologize for the confusion here. The self-consistency of measured data was checked using the Gibbs-Duhem equation, while the experimental points were fitted to self-consistent functions. This sentence was changed to:*

**Changes in manuscript (section 1, line 31): In addition, Hansen et al. (1955) represented the experimental points using self-consistent activity coefficient functions.**

• L. 32 – 34: The sentence should be improved given that, among the cited references, both Peng et al. (2001) and Choi and Chan (2002) use electrodynamic balance measurements that cover mass fractions of the solute far beyond the dilute solution range of the dicarboxylic acids. Further, the COSMOtherm predictions provided in this work (Fig. 3) seem not to be compared to such experimental data sets, even though the authors are aware of them.

*Author's response: We agree that the comparison with experiments should be more transparent. Measurements for supersaturated aerosol solutions were added to the activity coefficient comparisons. A figure of malonic acid and water activity coefficients compared to different experimental and UNIFAC values was moved to the main text. Comparison between experiments and COSMOtherm calculations of other compounds are shown in the Supplement. In addition, the text of section 2.3.2 was revised (see the response to the comment above on page 4-6).*

• L. 35: Define abbreviations for COSMOtherm and COSMO-RS.

*Author's response: Definition of COSMO-RS was added to line 35. COSMOtherm is a name of a program, not an abbreviation.*

• L. 70: Clarification necessary; Eq. (1), (2) define activity coefficients using the pseudo-chemical potential, which differs from the "regular" chemical potential more often used. However, comparison of Eqs. (1) and (2) raises the question how $\mu_{*o,li}$ in Eq. (1) differs from $\mu_{oi}$ in Eq. (2)? Eq. (2) seems to express the same relationship as Eq. (1). How exactly do they differ and what convention is used for the activity coefficients in Eq. (2)? Also, on line 71, the gas constant is expressed using kcal for energy; use of SI units would generally be preferred.

*Author's response: Eq (2) is the definition of pseudo-chemical potential, where the reference state is not specified. If the same reference state is used, $\mu_i^{*o}$ and $\mu_i^o$ are equal. We added the following to clarify the definition. All energy units were converted to kJ/mol.*

**Changes in manuscript (section 2.1, line 71): By definition, the activity coefficient of a compound at the reference state is unity ($\gamma_i^l(x_i=1)=1$), which leads to equal chemical and pseudo-chemical potential at the reference state. At other states ($x_i<1$), the relation between chemical and pseudo-chemical potentials ($\mu$ and $\mu^*$, respectively) can be expressed as**
$$\mu_i^*(x_i)=\mu_i(x_i)-RT\ln x_i$$

• L. 76 – 79: Clarify the basis for Eq. (3), why should that apply (references /reasons)? Also, Eq. (4) stated in the current form seems incorrect and awkward: why write the right hand side composition as $a_{I,\beta i}(1-x_{SOL,w})$? This seems not entirely correct and to be a potential source of confusion. This confusion exists because $(1-x_{SOL,w})$ should be exactly the same as $x_{SOL,acid}$ in a binary mixture(in the same phase), yet the former expression would only be correct for binary aqueous mixtures, not in general. However, in the LLE case, the mole fractions of acid in phases α and β will differ, which is missing in Eq. (4). Why not write $a_{I,\beta i}(x_{\beta SOL,acid})$ and analogously for phase α. Further, please define the meaning of subscript SOL.

*Author's response: The specific compounds were removed from the definition of LLE to make it more general and the derivation of Eq. (3) was added. Furthermore, the definition of $x_{SOL}$ was added.*

**Changes in manuscript (section 2.2, line 74): In LLE, the standard chemical potential (μ) of a compound is equal in both of the liquid phases (α and β):**

$$\mu_i(x_i^\alpha)=\mu_i(x_i^\beta)$$

**The standard chemical potential of compound i in a solution is defined using the standard chemical potential at the reference state:**

$$\mu_i(x_i)=\mu_i^o(x^o,T,P)+RTlna_i(x_i),$$

**where $a_i(x_i)$ is the activity of compound i with mole fraction $x_i$. The liquid-liquid equilibrium condition between the solvent-rich phase (α) and the solute-rich phase (β) becomes:**

$$a_i(x_i^\alpha)=a_i(x_i^\beta)$$

**Changes in manuscript (section 2.2, line 82): … where $x_{SOL,i}$ is the mole fraction solubility (SOL) of compound i in the solvent.**

• L. 98: The following statement "by multiplying the reaction equilibrium constant with the ratio of the activity coefficients..." and need for Eq. (11) seem unwarranted and require further explanation. Why should Eq. (11) be necessary? Is this because the authors are only considering mole fractions in the reaction constant, not activities? It is unclear because based on Eq. (8) – (10), in which chemical potentials and therefore implied activity coefficients are used, there seem to be no need for Eq. (11). Are you instead using pseudo-chemical potentials? Please clarify this. Also, activity coefficients of the reactants and products in what mixture? If the mixture contains a solvent, e.g. water, which affects the values of activity coefficients, how would Eq. (11) become solvent-independent?

*Author's response: We agree that this should be formulated more clearly in the manuscript. The definition of effective equilibrium constant was rewritten without using the equilibrium constant.*

**Changes in manuscript (section 2.3, line 91): COSMO*therm* estimates effective equilibrium constants of condensed-phase reactions from the free energy of the reaction ($\Delta G_r^{lo}$):**

$$K_{eff} = exp(\Delta G_r^{lo}/RT)$$

**The reaction free energy is calculated from the free energies of the pure reactants ($G_{react}^{lo}$) and products ($G_{prod}^{lo}$):**

$$\Delta G_r^{lo} = \Sigma \; G_{prod}^{lo} - \Sigma \; G_{react}^{lo}$$

**The free energy of compound i is the sum of the energy of the solvated compound ($E_{COSMO}$), the averaged correction for the dielectric energy (dE; Klamt et al., 1998) and the pseudo-chemical potential of the pure compound:**

$$G_i^{lo} = E_{COSMO,i} + dE_i + \mu_i^{*,lo}(x^o,T,P)$$

• L. 114: "equilibrium constants on a mole fraction basis" – Unclear what "mole fraction basis" should imply here. If the equilibrium constant is computed based on free energies of reaction or chemical potentials of reaction, they are always dimensionless – as any thermodynamic equilibrium constant should always be (concentration-product-based equilibrium constants are only approximations and not thermodynamically correct).

*Author's response: We agree with the reviewer, the phrase "mole fraction basis" was removed from the manuscript.*

• L. 130 – 133: Eq. (15) and text: It is unclear why enthalpic and entropic energy contributions should not already be accounted for by $\Delta_0$. Also why is there a factor 2? It seems possible that the energy difference is just not known/predicted well enough, such that a fit parameter was introduced to match experimental data. Is that the motivation for the "enthalpic" contribution in Eq. (15)? Please discuss.

*Author's response: These fitting parameters were used since the COSMOtherm program cannot determine the energy of the monomers in the clusters. The calculation of the interaction energy has been further clarified in the manuscript.*

**Changes in manuscript (section 2.4, line 128): The formation free energy of the cluster (G(A,A·B)) is calculated using fitting parameters $c_H$ and $c_S$ (enthalpic and entropic contributions, respectively) to describe the interaction between the monomers A and B in the cluster (A·B):**

$$G(A,A \cdot B) = c_H - c_S T,$$

**The fitting parameters are used because COSMO*therm* is unable to calculate the energy of a monomer in a cluster. Instead, the energy of a monomer in a cluster is assumed to be equal to the energy of the lowest-energy conformer of the same compound and the favorability of the cluster formation is estimated using the fitting parameters. Without temperature dependent experimental data, it is not possible to fit both fitting parameters. We therefore consider the enthalpic parameter $c_H$ as the total formation free energy parameter at 298.15 K, setting the entropic parameter $c_S$ to zero.**

• L. 180 – 184: The discussion of hydration and dimerization in aqueous solutions raises the question whether the dissociation of the carboxylic acids into dissolved ions was considered in the simulation? In dilute aqueous solution, dissociation of the acid and formation of hydronium ions would seem to be favorable over non-dissociative hydration. Please discuss. Acidity and pH are mentioned in the introduction, but nothing is said about acid dissociation within COSMO-RS-DARE simulations.

*Author's response: We have added a comparison between activity coefficients of dissociated and non-dissociated oxalic acid (lowest pK$_a$) to the Supplement. For further details see the response to the first comment above (page 2-3).*

• L. 198 – 199: Statement: "We used these experimental activity coefficients to fit the enthalpic parameters (c$_H$) for each of the acids in the COSMO-RS-DARE calculations." Given that such fit parameters were introduced, how predictive is the outlined COSMO-RS-DARE method for calculation of activity coefficients in aqueous solutions for compounds where the experimental equilibrium constants or activity coefficients are not known? Elaborating the discussion on this key point seems important for applications in atmospheric aerosol modeling. For example, if these parameters were all ignored (set to zero), how different would the predictions be for activity coefficients and solubilities in water?

*Author's response: The manuscript text was revised to emphasize that fitting parameters are only needed to improve the activity coefficient and solubility calculations of monocarboxylic acids and dicarboxylic acids are described well by COSMOtherm without the COSMO-RS-DARE extension.*

**Changes in manuscript (section 3.2.1, line 216): If the enthalpic parameters in COSMO-RS-DARE calculations are not fitted to experimental activity coefficients and instead are set to zero, the activity coefficients of both acid and water underestimate the experimental activity coefficients of Hansen et al. (1955) (see Fig. S2 of the Supplement).**
**(section 3.2.1, line 221): While COSMO-RS is fully predictive, COSMO-RS-DARE requires parameter fitting using experimental data. Fitted COSMO-RS-DARE parameters from one system can be used in other systems where the same clustering reactions are relevant. For instance, Sachsenhauser et al. (2014) found that the same interaction parameters of acid dimers can be used in systems containing other similar (non-polar) solvents. This indicates that our interaction enthalpies can be applied to other aqueous systems, for instance, ternary systems containing an inorganic salt, in addition to the carboxylic acid and water. This would allow for extending the findings of this study to atmospherically relevant aerosol solutions.**
**(section 3.2.1, line 216): If no experimental activity coefficients are available for fitting the COSMO-RS-DARE parameters, COSMO-RS estimates agree with experiments overall better than COSMO-RS-DARE or UNIFAC. COSMO-RS-estimated acid activity coefficients are close to the measured values in all mixing states, and for water activity coefficients the agreement between COSMO-RS and experiments is good in mixing states with x$_{acid}$<0.75.**
**(section 4, line 307): We were also able to estimate activity coefficients of pentanoic and hexanoic acids using only experimental water activity coefficients in the fitting of the COSMO-RS-DARE enthalpic parameters.**

**Changes in Supplement:**

[Figure]

**Figure S2: Activity coefficients ($\gamma^l$) of (a) monocarboxylic acids and (b) water in all mixing states of the binary aqueous solutions, at 298.15 K. The solid lines represent activity coefficient estimates using COSMO-RS-DARE ($c_H$=0), dashed lines are UNIFAC estimates, dotted lines are calculated from the equations fitted to experiments by Hansen et al. (1955), and the markers are the experimental points from the same study.**

• L. 214, 215: Do these calculation include phase separation for the larger acids or is a mixed phase assumed for all compositions? At higher organic acid mole fractions, the binary mixtures show high values for water activity coefficients, as also stated in the caption to Fig. 2. This may suggest liquid-liquid phase separation could occur, which could affect the interpretation of the experimental data used for comparison. Please discuss.

*Author's response: Phase separation occurs in aqueous pentanoic and hexanoic acid mixtures based on experimental solubilities and COSMOtherm calculations. The experimental points by Hansen et al. (1955) shown in Fig. 5 for water activity coefficients in pentanoic and hexanoic acid were measured in mixing states corresponding to the acid-rich phases.*

**Changes in manuscript (Figure 2 caption): For the studied acids with finite aqueous solubilities at 298.15 K (pentanoic and hexanoic acid), water activity coefficients were measured using acid-rich solutions (Hansen et al., 1955).**

• L. 261: "However, without experimental activity coefficient data of oxalic acid–water systems, we are not able to fit the $c_H$ parameter needed for reliable estimates." – This statement is simply not correct. A literature search reveals that there exist multiple useful measurements for this binary system, such as water activities at dilute and concentrated conditions, from several sources. These include the work of Maffia and

Meirelles (2001) mentioned earlier in the study, as well as work by Braban et al. (2003) and Marsh et al. (2017), which provide water activity data that should be used here.

*Author's response: We highly appreciate the supplied references and a comparison of water activity coefficients in oxalic acid were added to the Supplement.*

**Changes in Supplement:**

[Figure]

**Figure S5: COSMO*therm*-estimated water activity coefficients in aqueous (a-b) oxalic acid, (c) adipic acid and (d) pimelic acid solutions at 298.15 K. The experimental and model activity coefficients are by Peng et al.,[S6] Marsh et al.,[S8] Maffia and Mereilles,[S10] Braban et al.[S11] and AIOMFAC-web.[S3]**

• L. 264 and Fig. 3: The partial dissociation of dicarboxylic acids is either not considered at low acid concentrations or not discussed, even though dissociation would seem to be likely, especially for the smaller acid molecules. Was it determined to be irrelevant? This will require adequate discussion. Also, mention that mole fractions used in this work are defined assuming undissociated acid molecules (if this is indeed the case).

*Author's response: We agree that it is necessary to further clarify whether dissociation is taken into account. We added a comparison between activity coefficients in non-dissociated and dissociated oxalic acid-water systems, see response to the first general comment of referee #1 above (page 2-3). In addition, the following sentence was added to the manuscript:*

**Changes in manuscript (section 2.1, line 71): Unless otherwise mentioned, the mole fractions $x_i$ correspond to mole fractions of undissociated acid or neutral non-protonated water.**

• L. 290 – 294: On melting temperatures and solids considered:"Cornils and Lappe (2000) and Omar and Ulrich (2006) also measured the melting point of oxalic acid and

found temperatures almost 100 K higher than Booth et al. (2010). ..." In this context, are the differences in melting points due to unclear statements about the crystalline form of oxalic acid (anhydrous or dihydrate)? Omar and Ulrich (2006) show in their paper that the solid–solid phase transition from the dihydrate to the anhydrous oxalic acid crystal polymorph occurs around 378 K, which is close to the melting temperature stated by Booth et al., while the anhydrous oxalic acid melts at 465 K. Therefore, the correct values of use in the COSMOtherm simulations will depend on the temperature range of interest; for room temperature, using the dihydrate from is very likely the crystalline solid to be considered and consequently the solubility equilibrium should be solved for oxalic acid dihydrate not the anhydrous form. It is also clear that the actual melting temperature and enthalpy of fusion is not as uncertain as the current text implies. Using the correct equilibrium relations for the dihydrate the agreement between COSMOtherm and the measured solubility data would be expected to be much better than shown in Fig. 3.

*Author's response: The melting point reported by Booth et al. (2010) was removed and the lower limit free energy of fusion was calculated using the melting point measured by Cornils and Lappe (2000) instead. In COSMOtherm, the SLE is an equilibrium between anhydrous solute and solvent. The transition considered for the free energy of fusion calculation is from anhydrous solid into anhydrous liquid.*

**Changes in manuscript (Table 1 caption): The melting point of oxalic acid reported by Booth et al. (2010) (370 K) is likely the temperature of the phase transition from dihydrate to anhydrous crystal polymorph. Similar transition was seen by Omar and Ulrich (2006) at 378.35 K in their differential scanning calorimetry experiment.**

• L. 305: "to significantly improve the activity coefficient estimates ..." – compared to what? COSMO-RS or other methods? What about a comparison to methods like UNIFAC, such as the model by Peng et al. (2001).

*Author's response: UNIFAC estimates by Peng et al. (2001) were added to the activity coefficient comparison figures (Figs 5, S1, S3-S5).*

**Changes in manuscript (section 4, line 305): We compared COSMO*therm*-estimated activity coefficients and aqueous solubilities of simple carboxylic acids with experimental values and a commonly used UNIFAC model, and found a good agreement between experiments and COSMO*therm* estimates. Using COSMO-RS-DARE, we were additionally able to improve the agreement between estimated and experimental water activity coefficient in binary monocarboxylic acid-water systems significantly compared to using COSMO-RS or UNIFAC.**

• L. 308: "In addition, COSMO-RS-DARE is able to predict the miscibility of butanoic acid in water, while COSMO-RS predicts a finite solubility" – This is nice. However, how much of that success comes from fitting model parameters (cH) rather than the use of the DARE extension?

*Author's response: The success of using COSMO-RS-DARE comes from both fitting the interaction parameters and selecting the appropriate clustering reactions for each calculation. Here, we have fitted the enthalpic parameter using activity coefficients of water and acid in the binary systems and used those parameters to estimate liquid-liquid equilibria. Fitting parameters are needed in the COSMO-RS-DARE method, otherwise the favorability of clusters formation is not modeled correctly. Similarly,*

*COSMO-RS and group contribution methods are parametrized using experimental data. The difference between COSMO-RS-DARE and COSMO-RS is that COSMO-RS-DARE is not (yet) parametrized in COSMOtherm and the user needs to fit the parameters for each compound.*

**Changes in manuscript (section 4, line 307): In addition, COSMO-RS-DARE was able to predict the miscibility of butanoic acid in water (using the fitting parameters of activity coefficient calculations), while COSMO-RS predicted a finite solubility.**

• Table 1: State whether the listed data are for the pure anhydrous solids of the acids or for hydrates (especially in case of oxalic acid). Also replace "literature values" by more appropriate wording.

*Author's response: We agree that the form of the solid should be clarified in the table. Furthermore, we have changed the formulation "literature values" to "experimental melting points".*

**Changes in manuscript (Table 1 caption): List of the studied α,ω-dicarboxylic acids and their experimentally determined melting points and heats of fusion. The values were measured using anhydrous acids (crystalline at 298.15 K).**

• Table 2: state the temperature for the listed equilibrium constants and for completeness also for which phase / solvent they apply (given that there is also gas-phase dimerization of such acids).

**Changes in manuscript (Table 2 caption): Dimensionless effective equilibrium constants ($K_{eff}$) of cluster formation in condensed phase, at 298.15 K.**

• Figure 3: Why are the prediction data cut of at the solubility limits? It would seem useful to mark the solubility limit at 298.15 K for each acid, but to also show the predictions for the supersaturated range (which may apply in aerosols). This would also facilitate an extended comparison with experimental data existing for those higher concentrations, e.g. by Choi and Chan (2002) and Marsh et al. (2017). With the chosen log-scale for the x-axis, too much emphasis is put on the very dilute concentration range below $10^{-3}$ $x_{acid}$, which seems not to be insightful.

*Author's response: To further improve clarity, the activity coefficients in Figure 3 were extended to the whole mole fraction range and the x-axis was changed to linear scale. Using linear scale on the x-axis, only solubilities of malonic and glutaric acid would be visible in the figure. We have therefore decided not to show the solubilities of the acids in this figure.*

**Updated figure in the main text (Figure 3):**

[Figure]

**Figure 6. COSMO-RS- (solid lines) and UNIFAC-estimated (dotted lines; AIOMFAC-web, 2020) activity coefficients (γ$^I$) of (a) dicarboxylic acids and (b) water in the binary acid-water mixtures at 298.15 K. All COSMO*therm*-estimated activity coefficient values are given in Tables S5 and S6 of the Supplement.**

**2 References mentioned**
Braban, C. F., Carroll, M. F., Styler, S. A., and Abbatt, J. P. D.: Phase Transitions of Malonic and Oxalic Acid Aerosols, The Journal of Physical Chemistry A, 107,6594-6602, 10.1021/jp034483f, 2003.
Choi, M. Y., and Chan, C. K.: Continuous Measurements of the Water Activities of Aqueous Droplets of Water-Soluble Organic Compounds, The Journal of Physical Chemistry A, 106, 4566-4572, 10.1021/jp013875o, 2002.
Maffia, M. C., and Meirelles, A. J. A.: Water Activity and pH in Aqueous Poly-carboxylic Acid Systems, Journal of Chemical & Engineering Data, 46, 582-587, 10.1021/je0002890, 2001.
Marsh, A., Miles, R. E. H., Rovelli, G., Cowling, A. G., Nandy, L., Dutcher, C. S., and Reid, J. P.: Influence of organic compound functionality on aerosol hygroscopicity: dicarboxylic acids, alkyl-substituents, sugars and amino acids, Atmos. Chem. Phys.,17, 5583-5599, 10.5194/acp-17-5583-2017, 2017

**Anonymous Referee #2**

General comment
This paper employs the recently developed COSMO-RS-DARE model to estimate activity coefficients and solubilities of carboxylic acids in water. COSMO-RS-DARE is an extension of COSMOtherm that takes dimerization and aggregation in solution explicitly into account. This technical note concludes that COSMO-RS-DARE leads to better agreement with experimental data than COSMOtherm for the investigated mixtures. Although this paper is submitted as a technical note, the technical description of COSMOtherm and its extension COSMO-RS-DARE is lacking a proper

derivation and explanation. Also, the benefit of COSMO-RS-DARE compared with COSMOtherm remains unclear. As it seems, the new method relies on experimentally determined activity coefficients to calculate dimerization equilibria. Therefore, the benefit of COSMO-RS-DARE in the absence of experimental data is unclear. It is not clear whether COSMO-RS-DARE just performs better in predicting solubilities because of an additional degree of freedom introduced through potential dimerizations or a more accurate description of the system.
Major revisions of the manuscript are required before this technical note can be considered for publication. The different COSMO versions need to be explained better and the discussion of the results needs to be improved.

*Author response: Detailed descriptions and derivations of both COSMO-RS and COSMO-RS-DARE have already been published in papers cited in the work. In this paper, we test these methods for an atmospherically relevant chemical system, repeating the derivation here would be beyond the scope of the paper. However, we have improved the general explanation of the methods in the context of testing the methods, as described in more detail above as a response to comments from referee #1. In addition, we have clarified the comparison of COSMO-RS-DARE and COSMO-RS with other approaches and discussed the predictiveness of COSMO-RS-DARE, as described above as a response to referee #1. COSMO-RS-DARE was added to the title of section 2.4 to distinguish the COSMO-RS-DARE theory from COSMO-RS.*

**Changes in manuscript (section 2.4 title): Concentration dependent reactions (COSMO-RS-DARE)**

Specific comments
Lines 20 – 22: Here, acidity is mentioned as highly relevant. But the approach used in this technical note totally neglects deprotonation of acids.

*Author's response: Additional investigation of acid dissociation was added for the most acidic of the studied carboxylic acids, oxalic acid. See the response to the first general comment of referee #1 (page 2-3).*

Line 25: activity data of carboxylic acid-water systems is abundant as exemplified by the studies mentioned just below this sentence and there are even more. Please revise this sentence.

*Author's response: We apologize a slight mix-up here. This was meant to say "acid activity coefficients". This was added to the text.*

**Changes in manuscript (section 1, line 25): However, the acid activity data of carboxylic acid-water systems is much scarcer.**

Lines 65 – 66: the meaning of a pseudo-chemical potential should be explained.

*Author's response: More explanation for pseudo-chemical potential was added.*

**Chances in manuscript (section 2.1, line 71): Pseudo-chemical potential (Ben-Naim, 1987) is an auxiliary quantity defined using the standard chemical potential at the reference state $\mu^o$:**

$$\mu_i^*(x_i) = \mu_i^o(x^o, T, P) + RT \ln \gamma_i(x_i)$$

**Pseudo-chemical potential has recently been used in molecular level solvation thermodynamics as a replacement to chemical potential (Sordo, 2015).**

Line 79: activity should be replaced by the activity coefficient in this equation.

*Author's response: The LLE condition used in the COSMOtherm calculations is that the activity of compound i is equal in both phases. The assumption is that there is no pure compound phase in the system. The equation was revised as a response to a comment from referee #1 (page 8).*

Line 81: This equation should be derived or a reference should be given.

*Author's response: The reference (Eckert and Klamt, 2019) was added to the equation.*

Line 96: How is the dielectric energy calculated or defined?

*Author's response: We have added a reference (Klamt et al. 1998) to the averaged correction to the dielectric energy.*

**Changes in manuscript (section 2.3, line 95): The free energy of compound i is the sum of the energy of the solvated compound (E$_{COSMO}$), the averaged correction for the  dielectric energy (dE; Klamt et al., 1998) and the pseudo-chemical potential of the pure compound:**

Line 96: The difference between the chemical potential and the pseudo chemical potential is not clearly made and not explained. Here, the same symbol is used to refer to the chemical potential that was used before for the pseudo chemical potential.

*Author's response: We agree that it should be more transparent which chemical potential we refer to. The definition of pseudo-chemical potential as a function of chemical potential was added to the manuscript. In addition, "chemical potential" was replaced by "pseudo-chemical potential" in appropriate places to clarify that pseudo-chemical potential is the one that is used in all calculations.*

Line 100: Equation (11) needs to be explained better.
Line 112: The derivation of Eq. (14) remains obscure. The equation rather seems to be a definition of the effective equilibrium constant than a derived equation.

*Author's response: The definition of effective equilibrium was simplified. See response to a comment of referee #1 (page 8-9).*

Line 116: How is the surface of a molecule defined? Either explain here or give a reference.

**Changes in manuscript (section 2.4, line 117): The surface is considered as an interface between a virtual conductor around the molecule and the cavity formed by the molecule (Klamt and Schüürmann, 1993).**

Line 125: what is a property calculation?

*Author's response: By "property calculation" we mean a calculation of any thermodynamic property in COSMOtherm.*

**Changes in manuscript (section 2.4, line 125): "property calculation" was changed to "COSMO*therm* calculation"**

Line 125: what is the screening charge density? A scheme might help to explain it.

*Author's response: We thank the referee for this suggestion, a scheme clarifying screening charge densities was added to the manuscript.*

**Changes in manuscript (section 2.4, line 117): Each surface segment has an area ([Å$^{-2}$]) and a screening charge density ($\sigma$[e Å$^{-2}$]).**

[Figure]

**Figure 1. The $\sigma$-surfaces of succinic acid and water conformers used in COSMO-RS and COSMO-RS-DARE calculations. The conformer distributions in COSMO-RS-DARE include parts of cluster $\sigma$-surfaces (in this example a hydrate cluster). Color coding of $\sigma$-surfaces: red = negative partial charge, blue = positive partial charge, green = neutral partial charge, grey = omitted $\sigma$-surface.**

Line 128 – 129: Why are interaction sites of molecule B not treated the same way?

*Author's response: Molecule B is also included in the COSMO-RS-DARE calculations, the text was mistakenly left out of the manuscript. This has been corrected.*

**Changes in manuscript (section 2.4, line 126): Similarly, the clustering product of molecule B is included in the calculation by omitting the $\sigma$-surface assigned to molecule A from the $\sigma$-surface of A·B. Examples of these partial $\sigma$-surfaces are shown on the right hand side of Fig. 1.**

Line 133 – 134: Why is the entropic parameter kept zero? This seems arbitrary. Please justify.

*Author's response: We use only the enthalpic contribution to describe the total energy contribution, because we do not have temperature dependent measurements to fit both enthalpic and entropic parameters. The same result would be achieved if we*

*gave the entropic parameter any value so that $c_H - c_S T$ equals the enthalpic parameter we found in the fitting. The enthalpic parameter is only valid in the temperature of our calculations (298.15 K), for other temperatures the entropic parameter must be fitted separately using temperature dependent measurements. This has been further clarified in the text.*

**Changes in manuscript (section 2.4, line 131): Without temperature dependent experimental data, it is not possible to fit the entropic parameter. We therefore consider the enthalpic parameter $c_H$ as the total formation free energy parameter at 298.15 K, setting the entropic parameter $c_S$ to zero.**

Lines 150 – 158: This section is difficult to understand. A scheme might help.

*Author's response: We agree that a scheme will aid the reader. The following figure has been added to the manuscript.*

**Changes in manuscript:**

[Figure]

**Figure 2. The formation of dicarboxylic acid hydrate conformers. Color coding: green = C, white = H, red = O.**

Lines 225 – 227: This finding questions the benefit of the method.

*Author's response: The method still has its benefits. When $x_{water} \rightarrow 0$, an important part of the description of water is removed from the calculation (water in hydrate), since the system contains no water and thus no hydrates or water dimers. The method can still be applied to estimate activity coefficients of water in solutions where $x_{acid} < 0.9$.*

**Changes in manuscript: However, when the hydrate and water dimer reactions are included, COSMO-RS-DARE is not able to predict realistic**

**activity coefficients for water at high mole fractions ($x_{acid} > 0.9$) of the acids. This is likely due to the low concentration of water in the binary solution, leading to errors in the description of the interactions between water molecules. Still, COSMO-RS-DARE estimates agree well with the experiments at least up to 0.9 mole fraction of the monocarboxylic acids. This is an improvement compared to the UNIFAC model, which fails to reproduce experimental water activity coefficients already at acid mole fractions above 0.25. At very high acid mole fractions ($x_{acid} > 0.95$), COSMO-RS-DARE predicts several orders of magnitude higher activity coefficients than what was seen in experiments.**

Lines 234: I would not refer to dicarboxylic acids as being of low aqueous solubility. Some dicarboxylic acids have a high solubility. Moreover, data well into the supersaturated range is available (e.g. in Soonsin et al., 2010). This sentence needs to be revised accordingly.

*Author's response: We agree with the review's comment that some dicarboxylic acids have high solubilities. This section of the manuscript was reformulated to include comparison with additional experimental activity coefficients and UNIFAC predictions (see response to a comment from referee #1 on pages 3-6).*

Line 247: Figs S2 and S3 should be moved to the main manuscript.
Line 255: Fig. S4 should be moved to the main manuscript.

*Author's response: Figure S2 was moved to the main manuscript as an example of a comparison with experiments and UNIFAC (AIOMFAC). The comparisons with glutaric, oxalic, succinic, adipic and pimelic acid show similar agreement, and we therefore left Figs S3-S5 in the Supplement.*

**Changes in manuscript:**

[Figure]

**Figure 5. Activity coefficients of (a-b) malonic acid and (c-d) water in the binary mixtures at 298.15 K calculated using different clustering reactions in the COSMO-RS-DARE calculation. As a comparison are activity coefficients of malonic acid by Davies and Thomas (1956) (at 298.15 K given in convention III) and Soonsin et al. (2010) (particle measurements at various temperatures given in convention I) and of water by Maffia and Mereilles (2001), Choi et al. (2002), Wise et al. (2003), Peng et al. (2001), Marsh et al. (2017), Braban et al. (2003) and AIOMFAC-web (2020).**

Line 264: The logarithmic plot is not very informative. Rather show the figures from the SI here.

*Author's response: The x-scale of this figure was changed to linear and the whole mixing range was plotted in the figure. See response to the last comment of referee #1 (page 14-15).*

**New references:**
**AIOMFAC-web: version 2.32, http://www.aiomfac.caltech.edu, 2020.**

**Ben-Naim, A.: Solvation Thermodynamics, Plenum Press, New York and London, 1987.**

**Braban, C. F., Carroll, M. F., Styler, S. A., and Abbatt, J. P. D.: Phase transitions of malonic and oxalic acid aerosols, J. Phys. Chem. A, 107,6594–6602, https://doi.org/10.1021/jp034483f, 2003.**

**Eckert, F. and Klamt, A.: COSMOthermReference Manual, version C30, Release 19, COSMOlogic GmbH & Co, KG.: Leverkusen, Germany, 2019.**

**Fredenslund, A., Jones, R. L., and Prausnitz, J. M.: Group-contribution estimation of activity coefficients in nonideal liquid mixtures, AIChE J., 21, 1086–1099, https://doi.org/10.1002/aic.690210607, 1975.**

**Klamt, A. and Schüürmann, G.: COSMO: a new approach to dielectric screening in solvents with explicit expressions for the screening energy and its gradient, J. Chem. Soc., Perkin Trans. 2, pp. 799–805, https://doi.org/10.1039/P29930000799, 1993.**

**Marsh, A., Miles, R. E. H., Rovelli, G., Cowling, A. G., Nandy, L., Dutcher, C. S., and Reid, J. P.: Influence of organic compound functionality on aerosol hygroscopicity: dicarboxylic acids, alkyl-substituents, sugars and amino acids, Atmos. Chem. Phys., 17, 5583, https://doi.org/10.5194/acp-17-5583-2017, 2017.**

**Sordo, J. Á.: Solvation thermodynamics: two formulations and some misunderstandings, RSC Adv., 5, 96 105–96 116, https://doi.org/10.1039/c5ra17305a, 2015.**

**Zuend, A., Marcolli, C., Luo, B. P., and Peter, T.: A thermodynamic model of mixed organic-inorganic aerosols to predict activity coefficients, Atmos. Chem. Phys., 8, 4559–4593, https://doi.org/10.5194/acp-8-4559-2008, 2008.**

Zuend, A., Marcolli, C., Booth, A. M., Lienhard, D. M., Soonsin, V., Krieger, U. K., Topping, D. O., McFiggans, G., Peter, T., and Seinfeld, J. H.: New and extended parameterization of the thermodynamic model AIOMFAC: calculation of activity coefficients for organic-inorganic mixtures containing carboxyl, hydroxyl, carbonyl, ether, ester, alkenyl, alkyl, and aromatic functional groups, Atmos. Chem. Phys., 11, 9155–9206, https://doi.org/10.5194/acp-11-9155-2011, 2011.

---

## Author Response (AR2)

We thank the referees for their additional comments. We have revised the manuscript following the referees' suggestions. You can find answers to the referee comments (*in italics*) below with additions to the manuscript and supplement text (**in bold**).

**Referee #1**

The authors have provided responses to the questions, comments and issues pointed out by both referees. They have carried out additional calculations and have modified the manuscript to account for suggested changes, where applicable. For this second round of reviews of this technical note, I will focus on the replies and manuscript sections with changes. Most of the issues raised in the first round have been addressed well. I have found a few issues to be further addressed before this manuscript is finalized for publication.

The line numbers used in the following comments are those from the revised manuscript (ms version 3).

line 300: Regarding the acid dissociation reaction (R2) notation for oxalic acid, it is a bit odd that you write the reaction using HA for the acid, given that oxalic acid is a diacid. It would be better to write the reaction as $H_2A + H_2O = HA^- + H_3O^+$ and perhaps also list the second dissociation reaction involving $HA^-$ (even if not considered by COSMO-RS-DARE). Further along these lines, from the main text alone it remains unclear whether both acid dissociation reactions were considered in the model or not; should be clarified.

*Author's response: Thank you for this suggestion. We have added the second deprotonation reaction to the equilibrium reaction (R2). We also mention in the text why only the first deprotonation was considered.*

**Changes in manuscript: In this case, the system contains neutral oxalic acid ($H_2A$) and water ($H_2O$), as well as singly or doubly deprotonated oxalic acid ($HA^-$ or $A^{2-}$, respectively) and hydronium ion ($H_3O^+$) according to the dissociation equilibrium**

$$H_2A + 2H_2O \rightleftharpoons HA^- + H_3O^+ + H_2O \rightleftharpoons A^{2-} + 2H_3O^+ \textbf{ (R2)}$$

**While both acid groups of oxalic acid can be deprotonated, here we consider only the first deprotonation, because the second dissociation constant of oxalic acid in water is higher (3.81; Rumble, 2018) than the first one (1.25; Rumble, 2018) and has a smaller effect on the equilibrium.**

section 3.2.2, line 282 and Supplement, Fig. S3 – S5: The experimental data shown in these figures and related interpretation in the text needs to be corrected. For example, in Fig. S3, 'growth' mode data from EDB experiments by Peng et al. (2001) and Choi and Chan (2002) are shown. At higher acid concentrations, under growth conditions (also listed as 'condensation' for Peng et al data), dicarboxylic acids are mostly in the effloresced/crystalline state. The measurement data do not reflect aqueous solutions and cannot be used to derive water activity coefficients. It seems the authors used those data points as if they were showing values for a single aqueous solution, which is clearly incorrect. Showing such growth data is confusing and pointless for the comparison in these figures. Have a look at Fig. 8 from Choi and Chan (2002). It is clear from that figure that any EDB growth mode data for $a_w < 0.83$ is for the effloresced glutaric acid case, for which the aqueous phase composition, if present, is simply unknown from these measurements. Furthermore, any data for water activities less than about 0.3 are also indicative of efflorescence of the acid; therefore these data cannot be used to derive water activity coefficients. For reference, at a water activity of $\sim 0.4$, the mass fraction of solute (mfs in the Fig. 8 of Choi and Chan) is about 0.9, which corresponds to about $x_{acid}$ of 0.55 for the 'evaporation' data points. The water activity coefficient there is about 0.89 (convention I). There is no (valid) water activity coefficient data in purely aqueous solution of glutaric acid that would suggest water activity coefficients to increase substantially at high $x_{acid}$ (say for $x_{acid} > 0.6$). Therefore, the experimental data shown in Fig. S3 (and similarly in S4, S5) need to be corrected (no growth / condensation data fro EDB measurements should be shown, nor the evaporation branch data for low water activity < 0.3, depending on the acid). The related discussion in the main text and supplement needs to be corrected as there is no measurement data supporting an increase in water activity coefficient to large values at high $x_{acid}$ for the diacids. In Fig. S3d, the shown COSMO-RS, Acid dimer variant, UNIFAC and AIOMFAC model curves are likely showing the correct trend towards high acid fraction.

*Author's response: Thank you for bringing this to our attention. We have removed the experimental points from particle growth and condensation measurements from Figs 5, S3, S4 and S5.*

50 **Changes in manuscript (Figure 5):**

[Figure]

**(Figure S3):**

[Figure]

**(Figure S4):**

[Figure]

[Figure]

(Figure S5):

section 3.2.2, line 287: Correct the sentence; "supersaturated" and "crystalline" have different meaning and the phrasing here confuses this. An acid solution cannot be supersaturated and "crystalline" at the same time; in the presence of a crystalline phase, the remaining aqueous solution will automatically be saturated under equilibrium conditions (but not to be confused with the composition at the solubility limit).

*Author's response: This sentence was removed because the measurements of crystalline particles were removed from the comparison figures.*

Abstract, line 8: correct phrasing: "fitting parameters" should be "use of fit parameters". Also, this sentence would make more sense if it were stated for what purpose the fit parameters were introduced. Of course, with a sufficient number of fit parameters and associated functions, one could use all kinds of models to fit thermodynamic data.

*Author's response: The purpose of the fit parameters was added to the abstract.*
**Changes in manuscript: COSMO-RS-DARE is not fully predictive, but fit parameters found here for water–water and acid–water clustering interactions can be used to estimate thermodynamic properties of monocarboxylic acids in other aqueous solvents, such as salt solutions.**

line 89: Rephrase the following: "Pseudo-chemical potential has recently been used in molecular level solvation thermodynamics as a replacement to chemical potential". "replacement" is not a good description since the pseudo-chemical potential is not replacing the meaning or use of the chemical potential in thermodynamics; it simply expresses a different quantity, which

is why naming it "pseudo-chemical" remains an unfortunate choice by Ben-Naim.

*Author's response: This was corrected in the manuscript.*
**Changes in manuscript: Pseudo-chemical potential has recently been used in molecular level solvation thermody-**
**namics instead of chemical potential (Sordo, 2015).**

line 98 and Eq. (4): This sentence is unclear: "In LLE, the standard chemical potential ($\mu$) of a compound is equal in both of the liquid phases..." and also on line 101: "The standard chemical potential of compound $i$ in a solution is defined using the standard chemical potential at the reference state.". In both sentences, the authors seem to confuse the chemical potential with the term "standard chemical potential" – they are equivalent. The word "standard" has very specific meaning in thermodynamics and should not be confused with "regular" or "usual". In the second sentence, it should be "The standard chemical potential of compound $i$ in a solution is defined using the chemical potential at the reference state" (where, in convention I, the second term on the right hand side vanishes).

*Author's response: We decided to not use the term "standard" in this context to avoid misunderstandings. Additionally, the definition of LLE was edited based on a comment from Referee #2 (on line 139).*
**Changes in manuscript: In LLE, the chemical potential ($\mu$) of a compound is equal in both of the liquid phases ($\alpha$**
**and $\beta$).**

line 106, Eq. (6): The notation of this equation is wrong. The phase should be specified for each activity (e.g. as superscript; $a_i^\alpha$) since that is the point of the isoactivity condition among different phases. The same should also be corrected for Eq. (4). It is also potentially misleading to state the mole fraction in parenthesis, because this equation could be misunderstood as activity times mole fraction = ...

*Author's response: The phase was added to the superscript of activity and the mole fractions were removed to avoid misunderstandings.*
**Changes in manuscript:**
$\mu_i^\alpha = \mu_i^\beta$ **(4)**
$a_i^\alpha = a_i^\beta$ **(7)**

**Referee #2**

The authors have revised the manuscript according to the comments and suggestions of the reviewers. With these revisions, the manuscript is substantially improved and can be published subject to minor revisions outlined below:

Line 13: do you mean lack of feasibility and sample availability? If yes, the sentence needs to be reformulated.

**Changes in manuscript: This is highly beneficial for applications to atmospheric systems, as these data are typically**
**not available for a wide range of mixing states realized in the atmosphere, due to lack of either feasibility of the experi-**
**ments or of sample availability.**

Line 45: It is not clear to what "previously" refers in the new context of the revised manuscript.

*Author's response: This section was reformulated.*

**Changes in manuscript: Solubilities and activity coefficients of carboxylic acids have also been estimated using the COSMO-RS theory implemented in the COSMO*therm* program (COSMO*therm*, 2019). For instance, Schröder et al. (2010) estimated the aqueous solubilities of various polycarboxylic acids using the TZVP parametrization of COSMO*therm* and found that COSMO*therm* was able to predict the temperature dependence of the solubilities of dicarboxylic acids ($m = 2$–$8$) well, while the absolute solubility estimates were not in a good agreement with experiments. Additionally, Michailoudi et al. (2020) estimated the activity coefficients of monocarboxylic acids with even number of carbon atoms ($n = 2, 4, 6, 8, 10, 12$) at infinite dilution.**

Lines 89 – 90: The benefit of working with pseudo-chemical potentials should be explained.

**Changes in manuscript: The benefit of pseudo-chemical potential is that it is valid for any concentration and fluid mixture, while the conventional chemical potential cannot necessarily be used to describe infinite dilution ($x_i \rightarrow 0$)(Ben-Naim, 1978).**

Line 100: Why do you use in this equation the standard chemical potential and not the pseudo-chemical potential?

*Author's response: We changed the equations to use pseudo-chemical potential instead of chemical potential.*

**Changes in manuscript: Combining equations 2 and 3 gives the relation between chemical potential and pseudo-chemical potential at the reference state:**

$$\mu_i(x_i) = \mu_i^{*\circ}(x^\circ, T, P) + RT \ln x_i + RT \ln \gamma_i(x_i) \text{ (5)}$$

**Equation 5 can be substituted for chemical potential in equation 4, giving**

$$\mu_i^{*\circ}(x^\circ, T, P) + RT \ln a_i^\alpha = \mu_i^{*\circ}(x^\circ, T, P) + RT \ln a_i^\beta, \text{ (6)}$$

**where $a_i^\alpha$ and $a_i^\beta$ are the activities ($a = x\gamma$) of compound $i$ in phases $\alpha$ and $\beta$, respectively.**

Line 279: the convention III is explained in the supplement but not in the main text. Either an explanation should be added to the main text or there should be a reference to the supplement here.

*Author's response: We added a sentence about convention III in the manuscript with a reference to the Supplement.*

**Changes in manuscript: In convention III, acid activity coefficients are given with respect to a $1 \, \mathrm{mol \, kg^{-1}}$ solution reference state (see Supplement for more information).**

Lines 282 – 299: In this discussion, "COSMOtherm" is sometimes used to refer to COSMO-RS and COSMO-RS-DARE and sometimes only to refer to one of these implementations. This is confusing. E.g. line 286: what is meant here with both models? UNIFAC and COSMOtherm or COSMO-RS and COSMO-RS-DARE? Line 293: "with COSMOtherm slightly overestimating the experiments": Do you mean here COSMO-RS and COSMO-RS-DARE or just one of these? Line 297 – 299: do you mean just COSMO-RS or also COSMO-RS-DARE? The authors should consider to only use "COSMOtherm" when both implementations (COSMO-RS and COSMO-RS-DARE) are meant.

*Author's response: The methods were clarified in this section.*

**Changes in manuscript: We compared COSMO*therm*-estimated activity coefficients and aqueous solubilities of simple carboxylic acids with experimental values and a commonly used UNIFAC model, and generally found a good agreement between experiments and COSMO-RS estimates. Using COSMO-RS-DARE, we were able to further improve the agreement between estimated and experimental water activity coefficient in binary monocarboxylic acid–water systems significantly compared to using COSMO-RS or UNIFAC. The COSMO-RS estimates of monocarboxylic acid activity**

coefficient in aqueous solutions agree with the experiments quite well, and were further improved by COSMO-RS-DARE when the enthalpic fitting parameters were fitted using experimental activity coefficients. We were also able to estimate activity coefficients of pentanoic and hexanoic acids using only experimental water activity coefficients in the fitting of the COSMO-RS-DARE enthalpic parameters. In addition, COSMO-RS-DARE was able to predict the miscibility of butanoic acid in water (using the fitting parameters of activity coefficient calculations), while COSMO-RS predicted a finite solubility. However, in aqueous solubility calculations of pentanoic and hexanoic acid, COSMO-RS led to a better agreement between the experiments and estimates compared to COSMO-RS-DARE.

For dicarboxylic acid–water systems, COSMO-RS produced better agreement with experiments than COSMO-RS-DARE. The experimental water activity coefficients from different sources have large variations and COSMO-RS-estimated water activity coefficients fit within the range of experimental water activity coefficients obtained from bulk and evaporation measurements. We also found a good agreement between COSMO-RS-estimated and experimental acid activity coefficients at all acid mole fractions.

Lines 300 – 309: here activity coefficients are calculated including dissociation of oxalic acid. However, it is not stated whether this is done for COSMO-RS or COSMO-RS-DARE. Also, the explanation of the dissociation calculation given in the supplement is obscure. Why do you assume a pH value of 7.0 of the solution? The oxalic acid solution should be acidic. As starting point of the calculation, the equilibrium equation should be formulated. In Fig. S6 only the dissociation of all confs are given as dashed lines. But what are all confs? What are the 0 H-bond confs?

*Author's response: These points were clarified in both the manuscript and the Supplement. $pK_a$ is only available in pure water, which means that the approximation made here is that there is very little of the acid in the solution. This approximation means that the dissociation correction is less accurate in concentrated solutions.*

**Changes in manuscript: Additionally, we computed activity coefficients of oxalic acid (the most acidic dicarboxylic acid of this study) with consideration of the first dissociation step for of oxalic acid included in the COSMO-RS calculation.**

**Changes in Supplement: Dissociation of an acid HA in water can be described using an equilibrium reaction**

$$HA + H_2O \rightleftharpoons A^- + H_3O^+ \quad (3)$$

**where the degree of dissociation is given according to the dissociation constants ($pK_a$). The mole fractions of the ionic compounds are calculated using the dissociation correction at different mole fractions of the undissociated acid ($x_{acid}$).**

**This approximation assumes that there is only a small concentration of the acid in water so the pH of the solution is close to that of pure water. This assumption is made because the $pK_a$ value of the acid is only given for pure water, not acid water solutions. The predicted dissociation correction is therefore less accurate at high concentrations of acid than in low acid concentrations.**

**(caption of Figure S6): Activity coefficients were calculated using either the full set of acid conformers (all confs) or only conformers containing no intramolecular H-bonds (0 H-bond confs).**

**References**

[revised manuscript text omitted]